# QUERY-LEVEL UNCERTAINTY IN LARGE LANGUAGE MODELS

**Lihu Chen[1], Gerard de Melo[2], Fabian M. Suchanek[3], Gaël Varoquaux[4]**
[1]Imperial College London, United Kingdom
[2]Hasso Plattner Institute / University of Potsdam, Potsdam, Germany
[3]Telecom Paris, Institut Polytechnique de Paris, France
[4]Soda, Inria Saclay, France
`lihu.chen@imperial.ac.uk, gerard.demelo@hpi.de,`
`fabian.suchanek@telecom-paris.fr, gael.varoquaux@inria.fr`

## ABSTRACT

It is important for Large Language Models (LLMs) to be aware of the boundary of their knowledge, distinguishing queries they can confidently answer from those that lie beyond their capabilities. Such awareness enables models to perform adaptive inference, such as invoking retrieval-augmented generation (RAG), engaging in slow and deep thinking, or abstaining from answering when appropriate. These mechanisms are key to developing efficient and trustworthy AI. In this work, we propose a method to detect knowledge boundaries via **Query-Level Uncertainty**, which estimates if a model is capable of answering a given query *before* generating any tokens, thus avoiding the generation cost. To this end, we propose a novel, training-free method called **Internal Confidence**, which leverages self-evaluations across layers and tokens to provide a reliable signal of uncertainty. Empirical studies on both factual question answering and mathematical reasoning tasks demonstrate that our Internal Confidence outperforms several baselines in quality of confidence while being computationally cheaper. Furthermore, we demonstrate its benefits in adaptive inference settings, showing that for RAG and model cascading it reduces inference costs while preserving overall performance. The code is available at ⌂ `https://github.com/tigerchen52/query_level_uncertainty`

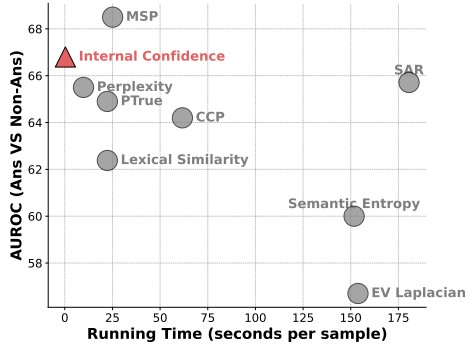

(a) Comparison of performance of distinguishing answerable and non-answerable queries and running time between our query-level Internal Confidence method and existing answer-level uncertainty measures (Qwen-14B on GSM8K).

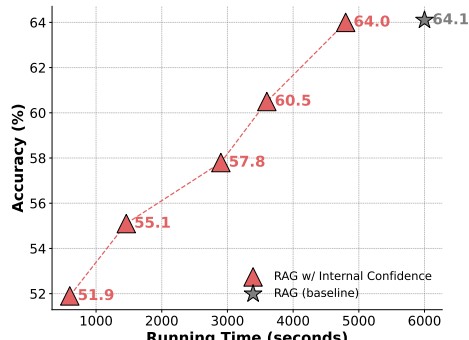

(b) Trade-off between running time and performance under different Internal Confidence thresholds for deciding on RAG invocation (Phi-3.8B on TriviaQA) compared against always using RAG.

Figure 1: Our *Internal Confidence* method improves performance / running time tradeoffs in factuality assessment and RAG settings.

# 1 INTRODUCTION

Large language Models (LLMs) have their knowledge boundaries (Li et al., 2024; Yin et al., 2024; Ren et al., 2025), which means that there are certain problems for which they cannot provide accurate answers. It is crucial for LLMs to be self-aware of their limitations, i.e., *to know what they know and know what they do not know* (Kadavath et al., 2022; Amayuelas et al., 2024).

Clear awareness of knowledge boundaries is central to improving AI, both for efficiency and trustworthiness. The rising usage of LLMs and agents has introduced significant computational and monetary costs (Varoquaux et al., 2025). For example, agentic workflows may cost 5×–25× more per query compared to a simpler LLM prompt (Anthropic, 2025). Regarding efficiency, if LLMs can distinguish answerable from non-answerable or simple from hard queries, they can smartly perform *adaptive inference* to navigate the trade-offs between computational cost and output quality (Chen & Varoquaux, 2024). For queries beyond their parametric knowledge, they can actively trigger RAG to obtain external knowledge (Lewis et al., 2020) or tool calls (Schick et al., 2023). When faced with hard problems, LLMs can engage in slow (or deep) thinking to improve their outputs, which is also known as test-time scaling (Snell et al., 2024; Zhang et al., 2025). Alternatively, they can defer a complex problem to a larger model via model cascading (Dohan et al., 2022; Gupta et al., 2024). This adaptive inference ensures efficient allocation of computational resources, reducing costs while maintaining performance, especially for agentic scenarios. Beyond efficiency, estimating whether a query is answerable also enhances honesty and trustworthiness of LLMs. When faced with highly uncertain queries, models can adopt an abstention strategy (Wen et al., 2024) to withhold potentially misleading responses, important in high-stakes domains like healthcare (Tomani et al., 2024).

In this work, we introduce the concept of *Query-Level Uncertainty* to estimate a model's knowledge with regard to a given query. The central research question here is: *Given a query, can we determine whether the model can address it before generating any tokens?* Most existing work focuses on answer-level uncertainty, which measures the uncertainty associated with a specific answer and is commonly used to assess the reliability of model outputs (Shorinwa et al., 2024; Vashurin et al., 2025). In contrast, our approach shifts from post-generation to pre-generation, measuring how confidently an LLM can solve a given query, prior to answer generation, as illustrated in Figure 2. This approach avoids the computational cost of generating potentially long answers.

Prior research has explored different strategies for uncertainty estimation. One line of work learns a probe of internal states to predict uncertainties of queries (Gottesman & Geva, 2024; Kossen et al., 2024). Another branch of work attempts to teach LLMs to explicitly express "I don't know" in their responses via fine-tuning methods (Amayuelas et al., 2024; Kapoor et al., 2024; Cohen et al., 2024; Zhang et al., 2024a). One common issue of these studies is that they require fine-tuning and training

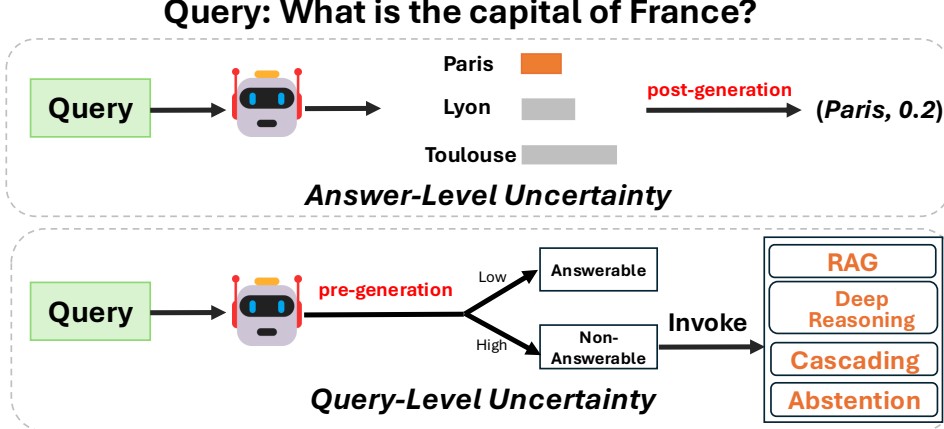

Figure 2: Illustrating the difference between answer-level and query-level uncertainty. Query-level uncertainty estimation distinguishes answerable from non-answerable queries (*knowledge boundary*) before generating answers, which is useful for adaptive inference, e.g., efficient RAG, fast–slow reasoning, or cascading models with different abilities.

samples, which introduces additional overhead and may restrict their generalizability across models and domains. To address this gap, we introduce a training-free approach to estimate query-level uncertainty that is both simple and effective.

Our training-free and generation-free approach, termed *Internal Confidence*, leverages self-evaluation across internal layers and tokens. It is grounded in a simple assumption: LLMs can internally self-assess the boundaries of their knowledge through a single forward pass over the given query, without generating an explicit answer. Inspired by the uncertainty measure P(TRUE) (Kadavath et al., 2022), we prompt LLMs with a yes–no question to self-assess if they are capable of answering a given query, and define the probability assigned to the token YES as the confidence level, denoted as P(YES). To fully exploit the latent knowledge within LLMs, our improved Internal Confidence approach computes this sort of P(YES) at each layer and token position. Subsequently, we aggregate these signals to obtain the overall confidence score. This aggregation is motivated by prior work showing that leveraging logical consistency across layers can improve outputs (Burns et al., 2022; Chuang et al., 2023; Xie et al., 2024). Concretely, we compute a weighted sum across layers and tokens, and the weights are derived from attenuated encoding (Chen et al., 2023), which enables fine-grained control of the influence of adjacent units.

To validate the effectiveness of our proposed Internal Confidence, we conduct experiments on three datasets that cover factual QA and mathematical reasoning tasks. For fair comparison, we adapt existing answer-level methods to the query level. Experimental results demonstrate that our proposed Internal Confidence can distinguish between answerable and non-answerable queries more accurately than a range of baselines, while being substantially faster than answer-level approaches (Figure 1a). In terms of applications, we showcase that our proposed method can support efficient RAG and model cascading. On the one hand, Internal Confidence can guide users to assess the trade-offs between cost and quality when invoking additional services. On the other hand, it reveals an "optimal point", where inference overhead can be reduced without compromising performance (Figure 1b). In conclusion, we introduce the notion of query-level uncertainty and propose a simple yet effective training-free method to estimate it, which enables models to determine whether a query can be addressed without generating any tokens.

## 2 RELATED WORK

### 2.1 UNCERTAINTY ESTIMATION AND LLMS

Existing approaches to LLM uncertainty primarily focus on estimating the uncertainty of LLM-generated responses, by providing a score intended to reflect the reliability of a query–answer pair (Geng et al., 2024; Shorinwa et al., 2024; Mahaut et al., 2024; Vashurin et al., 2025). These approaches often rely on internal states (Chen et al., 2024a) or textual responses (Kuhn et al., 2023), and commonly use calibration techniques to mitigate issues such as overconfidence (Zhang et al., 2024b) and biases (Chen et al., 2024b). Notably, these methods assess *post-generation* reliability, i.e., uncertainty regarding a specific answer after it has been produced. In contrast, relatively little work has explored how to quantify a model's ability to address a query prior to token generation. For example, Gottesman & Geva (2024) propose training a lightweight probe on internal representations to estimate the model's knowledge about specific entities. Similarly, Semantic Entropy Probes (Kossen et al., 2024) suggest that internal model states can implicitly encode semantic uncertainty, even before any output is generated. To the best of our knowledge, this work is the first to formally define query-level uncertainty and to investigate it systematically.

### 2.2 KNOWLEDGE BOUNDARY DETECTION

LLMs should be able to faithfully assess their level of confidence in answering a query. This awareness of knowledge boundaries (Li et al., 2024; Yin et al., 2024; Wang et al., 2024) is essential for building reliable AI systems, particularly in high-stakes domains such as healthcare and law. A pioneering study by Kadavath et al. (2022) explores whether language models can be trained to predict when they "know" the answer to a given query, introducing the concept of "I Know" (IK) prediction. Based on this idea, subsequent work has proposed methods to help LLMs become explicitly aware of their knowledge limitations through fine-tuning strategies (Amayuelas et al., 2024; Kapoor et al., 2024). Cohen et al. (2024) further advances this line of research by introducing a special [IDK] ("*I*

*don't know*") token into the model's vocabulary, allowing the direct expression of uncertainty in its output. Similarly, R-Tuning (Zhang et al., 2024a) tunes LLMs to refrain from responding to questions beyond their parametric knowledge. While these abstention-based approaches show benefits in mitigating hallucinations (Wen et al., 2024), they often require additional fine-tuning, which introduces overhead and may limit generalizability across models and tasks. In this work, we propose a training-free method to identify the knowledge boundary of an LLM, which offers a more efficient alternative that can be applied across models and tasks.

## 3 PROBLEM STATEMENT AND METHOD

In this section, we define the problem and introduce our *Internal Confidence*, a training-free and generation-free uncertainty that reflects whether an LLM can address a query in its own knowledge, prior to generating tokens.

### 3.1 PROBLEM STATEMENT

Given a query (including prompt tokens) $\mathbf{x} = (x_1, \ldots, x_N)$, we aim to quantify the query-level uncertainty, $U(\mathbf{x})$, without generating an answer $\mathbf{y}$. This differs from existing uncertainty approaches that estimate the uncertainty associated with a specific generated answer, an answer-level uncertainty that can be denoted as $U(\mathbf{x}, \mathbf{y})$. We define a query as being within the model's knowledge boundary if the LLM can produce a correct answer under greedy decoding, i.e., by selecting the highest-probability token at each step without sampling. Conversely, failure to produce the correct answer suggests the query falls beyond the model's boundary, and it does not possess sufficient knowledge to answer it. While greedy decoding ensures deterministic measurement, it may not always reflect the optimal performance of a model (Song et al., 2024), as alternative decoding strategies like beam search may elicit a better answer. We stick with greedy decoding for the following reasons: (1) Efficiency – Our method treats a successful greedy decode as a signal that the model knows how to answer, which is a fast proxy. In contrast, non-greedy decoding requires the configuration of beam numbers, probability thresholds, and sampling numbers, which complicates both the definition of knowledge boundary and the assessment cost. (2) Reproducibility – Greedy decoding outputs a single deterministic output for a given input, which offers a stable and reproducible baseline and benchmark.

Therefore, this pragmatic framework serves as a heuristic indicator of internal knowledge, rather than an absolute measure. We use this standard to evaluate the estimated query-level uncertainty, i.e., a lower uncertainty indicates a model is more likely to output the correct answer.

Our problem formulation mostly targets epistemic uncertainty of the model, though specific queries and datasets may contain aleatoric effects (see details in Section A), and the definition of the knowledge boundary is aligned with the *parametric knowledge boundary* (Li et al., 2024). This boundary of a model is the set of all knowledge encoded in its parameters that can be verified by at least one input–output pair. Our study focuses on queries with definite and clear-cut answers, as in factual QA and mathematical reasoning, which have broad applications and allow for clear evaluations. While contentious queries with open and subjective answers are also important in areas such as politics and philosophy, they remain beyond the scope of this work.

### 3.2 METHOD: FROM P(YES) TO INTERNAL CONFIDENCE

Studies have revealed that LLMs can express verbalized uncertainty in their responses (Tian et al., 2023; Xiong et al., 2024), and they can self-evaluate whether they know the answer to a question without reference to any specific proposed answer (Kadavath et al., 2022), which indicates that LLMs possess an internal mechanism for assessing the correctness of their outputs. At the same time, a recent work indicates that answerable and non-answerable questions are also linearly separable in hidden states (Slobodkin et al., 2023). Building on this observation, one can explicitly prompt an LLM to self-assess its confidence in answering a given query by constraining the response to a yes–no binary format: *"Respond only with 'Yes' or 'No' to indicate whether you are capable of answering the* {Query} *accurately. Answer Yes or No:"*. Following that, we can compute the

probability assigned to the token P(YES) at the last token ($x_N$):

$$P(\text{YES}) = \text{softmax}\left(\mathbf{W}^{\text{unemb}}_{[\text{YES,NO}]} \, \mathbf{h}_N^{(L)}\right)_{\text{YES}} \tag{1}$$

Here, $N$ is the index of the last token in the query and $L$ is the index of the last layer of the model. $\mathbf{h}_N^{(L)} \in \mathbb{R}^d$ is the hidden state, where $d$ is the dimensionality of the hidden representations. $\mathbf{W}^{\text{unemb}} \in \mathbb{R}^{|\mathcal{V}| \times d}$ is the unembedding matrix that maps the hidden state $\mathbf{h}_N^{(L)}$ to logits over the vocabulary $\mathcal{V}$. The unembedding layer provides meaningful and comparable probabilities, whereas the raw logits are not directly interpretable in this way. The probability P(YES) can serve as a query-level confidence score here, which is similar to the process of linear probing (Alain & Bengio, 2016), but without any training steps. While this measure correlates with verbalized uncertainty, a key distinction is that it requires only a single forward pass of the query, without generating any answer tokens.

However, P(YES) considers only the final hidden state of the LLM, although the intermediate internal states of LLMs preserve rich knowledge and latent information (Chen et al., 2025), especially for uncertainty estimation (Azaria & Mitchell, 2023; Chen et al., 2024a). Furthermore, prior work demonstrates that incorporating logical consistency across layers can improve outputs (Burns et al., 2022; Chuang et al., 2023; Xie et al., 2024).

Motivated by these insights, we propose the *Internal Confidence*, a method that leverages latent knowledge distributed across multiple layers and tokens. Formally, let $f_\theta$ denote the transformation function for computing hidden states, parametrized by $\theta$. The hidden state for the token $x_n$ of the input query at layer $l$ is computed as:

$$\mathbf{h}_n^{(l)} = f_\theta(\mathbf{h}_1^{(l-1)}, \dots, \mathbf{h}_n^{(l-1)}) \tag{2}$$

In total, the model contains $N \times L$ such latent representations, and we can use Equation 1 to compute the P(YES) for each $\mathbf{h}_n^{(l)}$.

Figure 3a plots the average P(YES) of Llama-8B on mathematical queries—the validation set of GSM8K (Cobbe et al., 2021)—across layers and query tokens.[1] We observe that the P(YES) generally increases from lower to higher layers and from left to right positions. If we treat each P(YES $\mid \mathbf{h}_n^{(l)}$) as a confidence score and compute the Area Under the Curve (AUROC), we can obtain an AUROC heatmap that illustrates how effectively each internal representation can distinguish answerable and non-answerable queries. As shown in Figure 3b, the highest score does not necessarily appear at the top right position. Instead, the representation $\mathbf{h}_5^{(27)}$ yields the best AUROC, and the performance gradually declines in regions surrounding this point. We refer to this optimal point as the *decision center*, where the model most effectively separates answerable from non-answerable queries.

To improve the vanilla P(YES), we can apply weighted average centering around the decision center, which serves as an ensemble strategy to enhance calibration and expressivity (Zhang et al., 2020; Stickland & Murray, 2020). We refer to this process as *Internal Confidence (IC)*, formally defined as:

$$\text{IC}(\mathbf{h}) = \sum_{n=1}^{N} \sum_{l=1}^{L} w_n^{(l)} \, \text{P}(\text{YES} \mid \mathbf{h}_n^{(l)}), \tag{3}$$

where $w_n^{(l)}$ denotes the weight assigned to the hidden representation $\mathbf{h}_n^{(l)}$. The equation describes a hierarchical two-step aggregation process. In the first step, for each individual token, we compute a weighted sum of confidence scores across layers. In the second step, we aggregate these token-level scores using another weighted average. Conceptually, this process can be parameterized by a layer weight vector $\mathbf{w}^{\text{layer}} \in \mathbb{R}^L$ for the first step and a token weight vector $\mathbf{w}^{\text{token}} \in \mathbb{R}^N$ for the second step. The obtained IC($\mathbf{h}$) value provides a single, refined confidence score that integrates rich information across both layers and tokens.

In our implementation, we adopt the top-right cell (corresponding to the last token and last layer) as the decision center, since we observe that the decision center tends to be located near the later layers

---

[1] Here, we consider the last $k$ tokens of a query, assuming that a model has seen the entire query and is able to infer its knowledge gap.

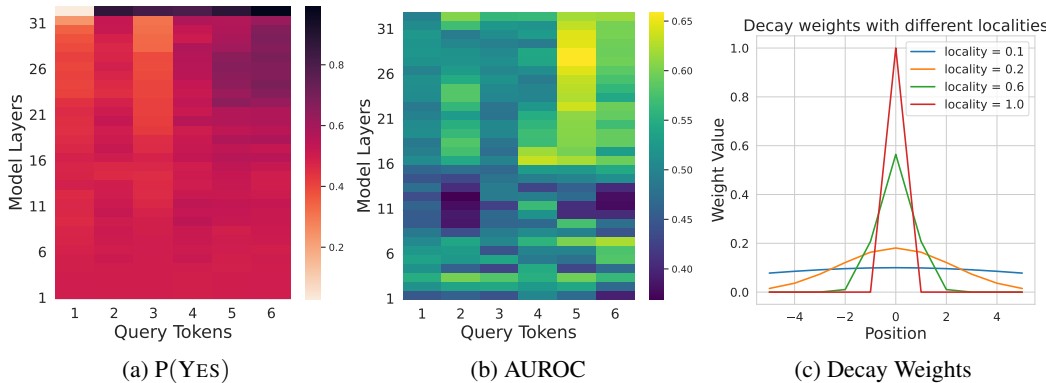

(a) P(YES)        (b) AUROC        (c) Decay Weights

Figure 3: **Left:** the internal P(YES) across tokens and layers. **Middle:** the AUROC of P(YES) across tokens and layers. **Right:** decay weights with different localities. Model: Llama-8B; Dataset: GSM8K validation set.

and final tokens across various architectures and tasks. While, in principle, the optimal decision center may also lie elsewhere, identifying such an optimal center would require a hold-out set of training data, which conflicts with our goal of developing a training-free approach. To address this, rather than relying on model- or task-specific tuning of the decision center, we incorporate information from the neighborhood of the fixed top-right cell. This strategy allows us to have the potential benefits of the optimal decision center while maintaining generalizability and avoiding dependence on additional training samples.

To reflect the observation that the AUROC performance gradually decays away from the decision center, we adopt Attenuated Encoding, as proposed by Chen et al. (2023), to compute the above weight vectors in Equation 3:

$$\delta_j^{(i)} = \frac{\exp(-\alpha \, |i - j|^2)}{\sum_{j=1}^{J} \exp(-\alpha \, |i - j|^2)}, \tag{4}$$

where $i$ is the index of the decision center, $|i - j|$ is the relative distance, and $\alpha > 0$ is a scalar parameter that controls the locality value. Locality is a metric that measures the extent to which weights are concentrated in adjacent positions of a center. Given a weight vector $\boldsymbol{\delta}^{(i)} = \{\delta_1^{(i)}, \delta_2^{(i)}, ..., \delta_J^{(i)}\}$ and assuming that the center index is $i$, we define its locality as

$$\text{Loc}(\boldsymbol{\delta}^{(i)}) \in [0, 1] = \sum_{j=1}^{J} \frac{\delta_j^{(i)}}{2^{|i-j|}}. \tag{5}$$

Here, a value of 1 implies that the vector perfectly satisfies the locality property, which means weights are extremely concentrated at the decision center. A low locality means weights are more uniformly assigned to neighborhoods. Figure 3c plots the weights obtained from Equation 4 for varying degrees of locality. This shows that we can account for the influence of neighboring layers and tokens during the averaging process.

Our proposed Internal Confidence is training-free and computationally efficient, as it requires only a single forward pass for a given query. Since model responses are frequently longer than input prompts and invoking external services such as RAG and deep thinking adds significant overhead, we propose this pre-generation uncertainty to support adaptive reasoning.

## 4 EXPERIMENTS

### 4.1 SETTINGS

**Models.** Our experiments consider three different LLM sizes: *Phi-3-mini-4k-instruct* (Abdin et al., 2024), *Llama-3.1-8B-Instruct* (Grattafiori et al., 2024), and *Qwen2.5-14B-Instruct* (Team, 2024).

| Method | TriviaQA ↑AUROC | ↑PRR | ↓ECE | SciQ ↑AUROC | ↑PRR | ↓ECE | GSM8K ↑AUROC | ↑PRR | ↓ECE | Avg ↑AUROC | ↑PRR | ↓ECE |
|---|---|---|---|---|---|---|---|---|---|---|---|---|
| *Phi-3.8B* | | | | | | | | | | | | |
| Max($-\log p$) | 55.5 | 10.0 | — | 51.4 | 2.9 | — | 55.0 | 11.3 | — | 54.0 | 8.1 | — |
| Predictive Entropy | 58.9 | 17.9 | — | 51.2 | 3.9 | — | **63.6** | **25.7** | — | 57.9 | 15.8 | — |
| Min-K Entropy | 59.9 | 20.0 | — | 52.7 | 4.9 | — | 60.4 | 17.9 | — | 57.7 | 14.3 | — |
| Attentional Entropy | 60.6 | 21.4 | — | 56.2 | 9.4 | — | 52.4 | 4.4 | — | 56.4 | 11.7 | — |
| Perplexity | 61.8 | 24.3 | — | 57.7 | 16.6 | — | 53.6 | 6.9 | — | 57.7 | 15.9 | — |
| Internal Semantic Similarity | 48.7 | -2.4 | 0.3 | 46.9 | -5.9 | 12.2 | 47.9 | -2.6 | 35.2 | 47.8 | -3.6 | 15.9 |
| P(YES) (*top right*) | **64.9** | 27.7 | **5.4** | **61.3** | 24.4 | 5.9 | 53.3 | 9.4 | 11.3 | **59.8** | 20.5 | **7.5** |
| P(YES) (*naive avg*) | 64.1 | 28.3 | 17.0 | 57.5 | 18.8 | 6.4 | 50.5 | 9.3 | 25.4 | 57.4 | 18.8 | 16.3 |
| Internal Confidence | 64.7 | **30.1** | 7.9 | 60.7 | **25.8** | 10.4 | 53.9 | 6.4 | 19.9 | **59.8** | **20.8** | 12.7 |
| *Llama-8B* | | | | | | | | | | | | |
| Max($-\log p$) | 54.9 | 11.1 | — | 51.4 | 1.9 | — | 53.3 | 10.4 | — | 53.2 | 7.8 | — |
| Predictive Entropy | 58.5 | 17.7 | — | 51.4 | 3.2 | — | **66.1** | 28.0 | — | 58.7 | 16.3 | — |
| Min-K Entropy | 58.1 | 17.4 | — | 53.5 | 7.9 | — | 57.5 | 13.2 | — | 56.4 | 12.8 | — |
| Attentional Entropy | 59.4 | 18.7 | — | 57.7 | 15.2 | — | 56.1 | 13.5 | — | 57.7 | 15.8 | — |
| Perplexity | 58.6 | 17.1 | — | 58.3 | 15.1 | — | 53.2 | 4.3 | — | 56.7 | 12.2 | — |
| Internal Semantic Similarity | 44.1 | -14.4 | 24.4 | 46.1 | -7.1 | 30.8 | 52.7 | 6.7 | 45.9 | 47.6 | -4.9 | 33.7 |
| P(YES) (*top right*) | 55.4 | 10.2 | 31.7 | **58.4** | **17.2** | 23.7 | 52.6 | 5.2 | 22.4 | 55.5 | 10.9 | 22.4 |
| P(YES) (*naive avg*) | 65.9 | 33.0 | 12.6 | 57.9 | 14.9 | 20.4 | 61.3 | 18.5 | 33.5 | 61.7 | 22.1 | 22.2 |
| Internal Confidence | **68.7** | **35.5** | 25.4 | 58.1 | 15.7 | **16.7** | 65.7 | **34.9** | **3.1** | **64.2** | **28.7** | **15.1** |
| *Qwen-14B* | | | | | | | | | | | | |
| Max($-\log p$) | 56.5 | 12.4 | — | 54.1 | 6.9 | — | 54.3 | 13.5 | — | 55.0 | 10.9 | — |
| Predictive Entropy | 59.3 | 18.9 | — | 53.2 | 6.9 | — | 66.4 | **32.6** | — | 59.6 | 19.5 | — |
| Min-K Entropy | 59.9 | 20.0 | — | 55.7 | 11.3 | — | 63.0 | 30.9 | — | 59.5 | 20.7 | — |
| Attentional Entropy | 59.1 | 17.2 | — | 59.4 | 19.2 | — | 54.9 | 3.1 | — | 57.8 | 13.2 | — |
| Perplexity | 59.1 | 17.8 | — | 60.1 | 20.7 | — | 54.0 | 7.3 | — | 57.7 | 15.3 | — |
| Internal Semantic Similarity | 51.0 | 2.5 | **2.0** | 45.5 | -7.7 | 14.9 | 47.5 | -4.6 | 33.1 | 48.0 | -3.3 | 16.7 |
| P(YES) (*top right*) | 67.8 | 36.0 | 30.3 | 60.0 | 21.7 | 24.1 | 55.0 | 11.7 | 6.4 | 60.9 | 23.1 | 20.3 |
| P(YES) (*naive avg*) | 67.0 | 33.9 | 3.5 | 59.5 | 17.9 | **14.6** | 64.0 | 32.3 | 32.4 | 63.5 | 28.0 | **16.8** |
| Internal Confidence | **71.9** | **43.3** | 26.5 | **62.6** | **23.6** | 18.2 | **66.8** | 28.2 | 5.7 | **67.1** | **31.7** | 16.8 |

Table 1: Overall results of different query-level uncertainty estimation methods. The best-performing methods are highlighted using boldface and second-best results are underlined.

This allows us to assess whether Internal Confidence generalizes across different model sizes. It is worth noting that Internal Confidence can also be applied to models without instruction tuning.

**Implementations.** For Llama and Qwen, Internal Confidence is computed in the zero-shot setting, whereas for Phi, we use two shots in the prompt, since smaller models benefit from demonstration-based guidance (See details in Section D.2). All LLMs employ greedy decoding to ensure deterministic outputs. The decision center is fixed to the last layer and last token, and we set $\alpha = 1.0$ (Equation 4) across all models and datasets.

**Evaluation Datasets.** We evaluate on two factual QA datasets and one mathematical reasoning dataset: TriviaQA (Joshi et al., 2017), SciQ (Welbl et al., 2017), and GSM8K (Cobbe et al., 2021). The first two tasks aim to assess factual knowledge stored in parameters, while GSM8K requires models to self-evaluate their reasoning capabilities. The ground truth for factual QA tasks takes the form of a short answer with entity-related facts. GSM8k as well calls for a short answer, but the intermediate reasoning steps are evaluated as well, following prior work (Kadavath et al., 2022). The three datasets consist of 10,000, 10,000, and 5,000 samples, respectively, with 1,000 samples from each reserved for validation.

We elicit responses from the model using a greedy decoding strategy. If the answer aligns with the ground truth, we consider the model as possessing sufficient knowledge and the query as falling within its knowledge boundary. For the first two datasets with short answers, answers are deemed correct if the ROUGE-L (Lin & Och, 2004) of the ground truth is greater than 0.3, which is consistent with prior work (Kuhn et al., 2023). For the GSM8K dataset, we use an LLM evaluator, Mistral-Large (MistralAI, 2024), to assess both reasoning steps and the final answer. We evaluate the reasoning steps on GSM8K because verifying the reasoning chain is essential to ensure the model truly understands the problem rather than outputting the correct results by chance. Subsequently, each query is paired with a binary label reflecting whether the model is capable of addressing it.

**Baselines.**  For comparison, we adapt state-of-the-art answer-level methods to quantify the pre-generation uncertainty (see details in Section C): (1) Max$(-\log p)$ (Manakul et al., 2023), (2) Predictive Entropy (Malinin & Gales, 2021), (3) Min-$K$ Entropy (Shi et al., 2024), (4) Attentional Entropy (Duan et al., 2024), (5) Perplexity, (6) Internal Semantic Similarity (Fomicheva et al., 2020), (7) P(YES) (*top right*), corresponding to Equation 1. (8) P(YES) (*naive avg*) is a variant of our Internal Confidence that adopts naive averaging to aggregate scores across different tokens and layers.

**Evaluation Metrics.**  We evaluate uncertainty by assessing whether a method can distinguish *answerable* and *non-answerable* queries, which can be treated as ranking problems, i.e., a lower uncertainty means a model is more likely to know the answer to the query. Following prior work (Manakul et al., 2023; Kuhn et al., 2023), we adopt the Area Under the Receiver Operating Characteristic Curve (AUROC) and Prediction Rejection Ratio (PRR) (Malinin et al., 2017) as metrics to measure this. Additionally, we compute the Expected Calibration Error (ECE) to assess the calibration of different methods.

## 4.2 INTERNAL CONFIDENCE CAN IDENTIFY ANSWERABLE AND NON-ANSWERABLE QUERIES

Table 1 summarizes the overall results comparing different query-level uncertainty methods. First, we can observe that our proposed Internal Confidence consistently outperforms other baselines in knowledge boundary detection, as reflected in both average AUROC and PRR. The advantage becomes more pronounced for larger models such as Llama-8B and Qwen-14B. For instance, on Qwen-14B, it obtains an average AUROC of 67.1 and PRR of 31.7, clearly surpassing all other methods. Regarding the calibration (ECE), Internal Confidence is found to consistently achieve a lower error across models and tasks. These findings indicate the effectiveness of Internal Confidence. Finally, we note that the variants, P(YES) (*top right*) and P(YES) (*naive avg*), generally underperform the full method, which highlights the importance of the attenuated encoding and its decay weights in effectively aggregating signals from different layers and tokens.

## 4.3 INTERNAL CONFIDENCE IS MUCH FASTER THAN ANSWER-LEVEL APPROACHES

We compare our query-level Internal Confidence with several popular answer-level uncertainty methods on GSM8K using Qwen-14B, including Perplexity (Fomicheva et al., 2020), Semantic Entropy (Kuhn et al., 2023), P(TRUE) (Kadavath et al., 2022), Lexical Similarity (Fomicheva et al., 2020), SAR (Duan et al., 2024), Maximum Sequence Probability (MSP), CCP (Fadeeva et al., 2024), and EV Laplacian (Lin et al., 2023).

Table 2 compares the effectiveness and run-time across different approaches. While answer-level approaches such as Perplexity, P(TRUE), and SAR require significantly higher computation time (ranging from nearly 10 seconds up to more than 180 seconds per sample), our Internal Confidence method achieves the best AUROC (66.8) with an average running time of only 0.3 seconds. This corresponds to speedups of over 30× to 600× compared to existing baselines. These results demonstrate that Internal Confidence achieves competitive performances compared to answer-level uncertainty approaches while being extremely faster, which can be a practical choice for tasks requiring longer and more complex answers.

| Method | ↑ AUROC | ↓ Time (s) | ↑ Speedup |
|---|---|---|---|
| Perplexity | 65.5 | 9.8 | 32× |
| Semantic Entropy | 60.0 | 151.8 | 506× |
| P(TRUE) | 65.2 | 22.3 | 74× |
| Lexical Similarity | 62.4 | 170.3 | 567× |
| SAR | 65.7 | 180.6 | 602× |
| MSP | **68.5** | 25.1 | 84× |
| CCP | 64.2 | 61.7 | 206× |
| EV Laplacian | 56.7 | 153.9 | 513× |
| Internal Confidence | 66.8 | **0.3** | — |

Table 2: Comparison with answer-level uncertainty methods (Qwen-14B on GSM8K).

Notably, the running time for Internal Confidence remains constant, independent of the length of answers. Figure 4 shows that the runtime of the best answer-level approach, SAR, grows with the answer length, reaching nearly 500s for answers over 600 characters. In contrast, Internal Confidence achieves large acceleration ratios, with speedups increasing as answers become longer, which demonstrates its scalability and efficiency. See results of other datasets in Table A1.

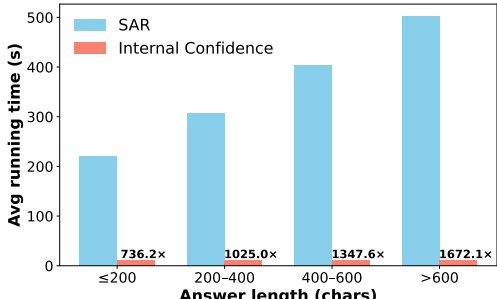 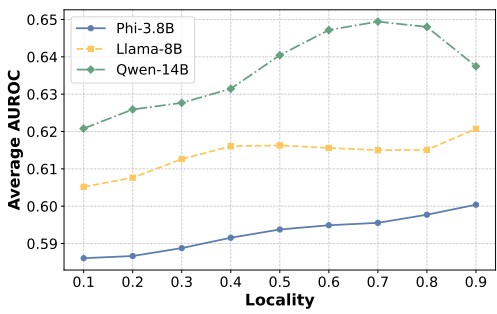

Figure 4: Acceleration ratio comparison between answer-level SAR and our Internal Confidence.

Figure 5: Impact of locality on validation set performance. We report the average AUROC across the three considered datasets. See details in Section D.3.

### 4.4 INTERNAL CONFIDENCE MAKES LLM REASONING MORE EFFICIENT

Recent studies advance LLM reasoning by introducing additional resources, such as using RAG to obtain external knowledge (Lewis et al., 2020) and inference-time scaling to improve outputs (Snell et al., 2024). However, it is not always necessary to use additional resources, especially for simple queries. Here, we use our proposed Internal Confidence for adaptive inference, determining when to invoke RAG, slow thinking, or model cascading.

We conduct experiments for two scenarios: (1) *Efficient (or adaptive)* RAG. This assumes that the Internal Confidence can serve as a signal of the knowledge gaps of a model. If the score is greater than a threshold, the model is sufficiently confident to address the query directly using its parametric knowledge. Otherwise, it requires invoking RAG. Existing studies have explore adaptive RAG through learned classifiers (Jeong et al., 2024; Marina et al., 2025) and answer-level uncertainty approaches (Jiang et al., 2023; Su et al., 2024; Yao et al., 2025; Moskvoretskii et al., 2025), which actively decide whether and when to retrieve documents. However, these approaches require training samples or generating answers to measure the uncertainty. In contrast, our Internal Confidence method is training-free and significantly faster than answer-level approaches (as shown in Table 2), which can serve as a potentially efficient way to guide adaptive RAG. We use the TriviaQA dataset for evaluation. This dataset provides web search results for a query, which can be used as retrieved contexts for RAG. (2) *Model Cascading.* This task aims to achieve cost-performance trade-offs by coordinating small and large models (Dohan et al., 2022; Gupta et al., 2024). The smaller models are responsible for easy assignments. Whenever they determine that the task exceeds their capabilities, it can be delegated to a larger model. We use a two-model cascade setting with Phi-3.8B and Llama-8B on the TriviaQA dataset. If the Internal Confidence of the smaller model is high, we do not invoke the larger model. Otherwise, the hard query is deferred to the larger model.

Figure 6 presents the results of applying Internal Confidence scores to efficient RAG (left) and model cascading (right). In both cases, the *trade-off region* illustrates how adjusting the confidence threshold allows us to balance efficiency and performance by controlling the frequency of external service calls or larger model invocations. The *optimal point* highlights thresholds where additional resource usage can be reduced without sacrificing accuracy. Results across the two tasks further confirm the effectiveness of Internal Confidence in identifying knowledge gaps. Our method offers practical benefits by reducing inference overhead, which can be applied to token-heavy agentic frameworks.

### 4.5 LOCALITY AFFECTS UNCERTAINTY PERFORMANCE

Our method incorporates attenuated encodings to aggregate probabilities centering around a decision point. The locality of the encoding may affect the accuracy of estimated uncertainties. To study the influence of the locality, we vary the $\alpha$ in Equation 4 to obtain encodings with different localities and observe how they affect the estimations. Figure 5 reports the average AUROC across three datasets and models. The results indicate that the effect of locality depends on both the task type and the

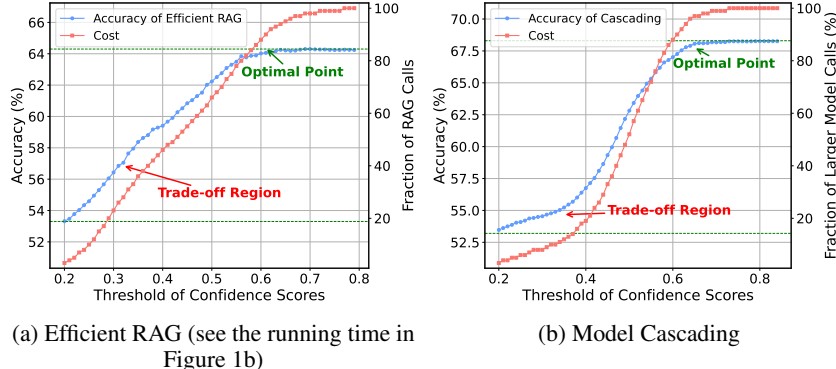

(a) Efficient RAG (see the running time in Figure 1b)

(b) Model Cascading

Figure 6: **Left:** We use estimated Internal Confidence to decide whether to invoke RAG. If the Internal Confidence exceeds a threshold, the model answers the query using its parametric knowledge. Otherwise, it relies on external knowledge. The plot shows the accuracy of Phi-3.8B on the TriviaQA dataset under this setting. **Right:** We implement a model cascading setting with Phi-3.8B (small) and Llama-8B (large) on the TriviaQA dataset. The Internal Confidence of the smaller model determines whether it answers the query or defers to the larger model when confidence is low. The green lines indicate the baseline accuracy achieved by the simple model or complex model.

model architecture. Although the optimal locality may vary with model and dataset (see details in Section D.3), we find that a default setting of $\alpha = 1.0$ (corresponding to Locality $\approx 0.7$) yields consistently competitive performance that generalize well.

## 5    CONCLUSION

In this work, we propose the new notion of query-level uncertainty, which seeks to assess whether a model can successfully address a query without generating any tokens. To this end, we propose the novel Internal Confidence technique, which leverages latent self-evaluation to identify the boundary of a model's knowledge. Extensive experimental results confirm the effectiveness of our approach on both factual QA and mathematical reasoning. Our method is capable of identifying knowledge gaps with a substantially faster speed compared to answer-level approaches. Furthermore, we apply Internal Confidence to two practical scenarios of adaptive inference, efficient RAG and model cascading. Our findings reveal that our method can identify two regions: a trade-off region and an optimal point. The former means that one can strike a balance between cost and quality by carefully selecting a threshold of confidence scores. The latter means that one can reduce inference overhead without compromising performance.

In conclusion, these results highlight Internal Confidence as a strong and general-purpose baseline for estimating query-level uncertainty. While there remains room for refinement, our study can serve as a strong baseline for this task, and we hope this study can stimulate future studies in this area.

## LIMITATIONS

There are several main limitations of this work. (1) Our proposed query-level uncertainty measure relies on access to a model's internal states, which is not feasible for fully black-box APIs. (2) We adopt straightforward fixed hyperparameters across all experiments for efficiency and generalizability, but this choice does not yield optimal performance in all settings. As discussed in Section D.5, our additional experiments show that the optimal decision center location varies across models and tasks. (3) Although internal confidence can serve as a strong baseline for detecting knowledge boundary, its performance still lags behind answer-level approaches. We hope this work inspires future research on more refined and robust ways to detect the knowledge boundary of foundation models.

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

# A    FUNDAMENTAL CONCEPTS

## A.1    ALEATORIC AND EPISTEMIC UNCERTAINTY

Uncertainty in machine learning is commonly categorized into two main types: aleatoric and epistemic uncertainty (Hora, 1996; Der Kiureghian & Ditlevsen, 2009; Hüllermeier & Waegeman, 2021). These distinctions are often overlooked in the context of LLM uncertainty estimation. Aleatoric uncertainty arises from inherent randomness in the data, such as ambiguous inputs or

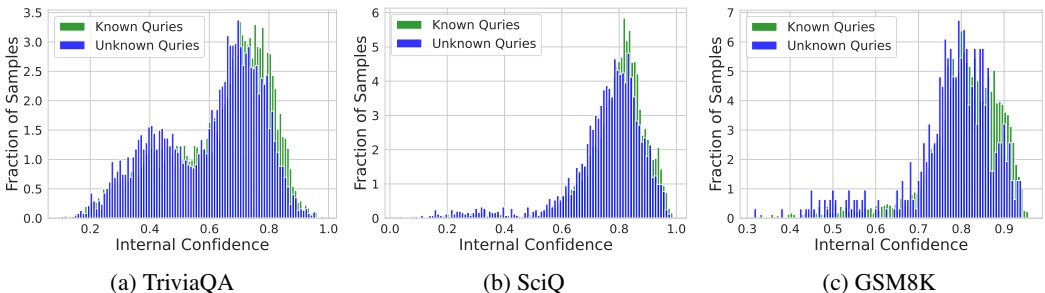

Figure A1: Distinguishing between answerable and non-answerable queries using Internal Confidence for Phi-3.8B.

conflicting annotations. This type of uncertainty is irreducible, as it reflects intrinsic noise in the input data. In contrast, epistemic uncertainty stems from a lack of knowledge, often due to insufficient training data and limited model capacity. Unlike aleatoric uncertainty, epistemic uncertainty is reducible with additional data or advanced modeling. In this work, we focus specifically on epistemic uncertainty, with the goal of evaluating whether an LLM possesses sufficient knowledge to answer a given query. For evaluation, we adopt factual QA and mathematical reasoning benchmarks, which are designed to have clear-cut answers. We assume these datasets are well-curated to minimize aleatoric uncertainty, such as ambiguous questions and inconsistent labels. However, we acknowledge that residual ambiguity may persist, given the inherent nature of linguistic ambiguity (Gillon, 1990) and the difficulty of fully disentangling aleatoric from epistemic uncertainty (Mucsányi et al., 2024). We treat such aleatoric effects as negligible for the purposes of focusing on epistemic uncertainty.

## A.2 UNCERTAINTY AND CONFIDENCE

In the context of LLMs, the terms uncertainty and confidence are often used interchangeably (as antonyms). However, the two concepts have subtle differences. As noted by Lin et al. (2023), uncertainty is a holistic property of the entire predictive distribution, while confidence refers to the model's estimated confidence level associated with a specific answer. For example, given a query $x =$ *"What is the capital of France"*, estimating uncertainty conceptually requires the distribution over all plausible answers, e.g., *Paris*, *Toulouse*, *Lyon*, etc., as operationalized by the semantic entropy framework (Kuhn et al., 2023), which clusters semantically equivalent outputs before computing entropy. In contrast, the conditional probability $P(Y = \text{Paris} \mid x)$ can serve as an indication of confidence here, reflecting how strongly the model supports that particular response. Given that it is unfeasible to enumerate all possible responses in our context of query-level uncertainty, we pragmatically treat uncertainty and confidence as antonyms.

## B PROMPT

We use the following prompt template for all experiments. The query $x$ consists of both prompt and question tokens.

*You are a helpful assistant that assesses whether you can provide an accurate response to a question. Respond only with 'Yes' or 'No' to indicate whether you are capable of answering the following question.* {Examples}{Input Question}.

## C BASELINE DETAILS

We adapt existing answer-level methods to quantify the pre-generation uncertainty, e.g., logit-based uncertainty. Given a query (including the prompt) $\mathbf{x} = (x_1, \ldots, x_N)$, we can obtain a probability for each token $P(x_n \mid x_{<n})$ by performing a forward pass. (1) The baseline $\text{Max}(-\log p)$ measures the query's uncertainty by assessing the least likely token in the query (Manakul et al., 2023). (2)

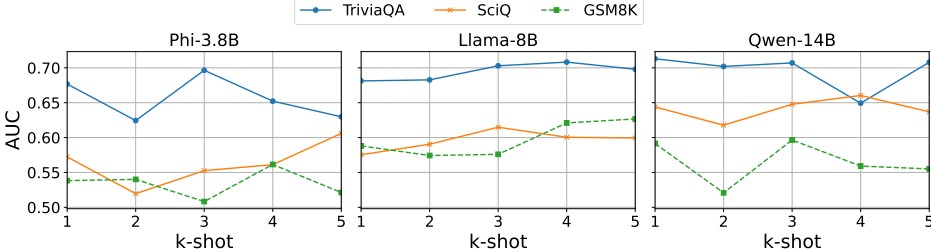

Figure A2: Impact of the number of in-context-learning example pairs on validation set performance.

*Predictive Entropy* is defined as the entropy over the entire query token sequence (Malinin & Gales, 2021):

$$\text{PE}(\mathbf{x}) = -\sum_{n=1}^{N} \log \text{P}(x_n \mid x_{<n}) \tag{A.1}$$

(3) *Min-K Entropy* combines the ideas of $\text{Max}(-\log p)$ and predictive entropy, by selecting the top-$K$ tokens from the query with the minimum token probability (Shi et al., 2024). (4) *Attentional Entropy* is a modified version of the predictive entropy that considers a weighted sum

$$\text{AE}(\mathbf{x}) = -\sum_{n=1}^{N} \alpha_n \log \text{P}(x_n \mid x_{<n}), \tag{A.2}$$

where $\alpha_n$ are the attentional weights for tokens $x_n$. The intuition here is that tokens contribute to the semantic meanings in different ways, such that we should not treat all tokens equally (Duan et al., 2024). (5) *Perplexity* reflects how uncertain a model is when predicting the next token:

$$\text{PPL} = \exp\left(-\frac{1}{N} \sum \log \text{P}(x_n \mid x_{<n})\right) \tag{A.3}$$

(6) *Internal Semantic Similarity* measures the average similarity among hidden states of different layers $\{\mathbf{h}_N^{(1)}, ..., \mathbf{h}_N^{(L)}\}$, which is inspired by lexical similarity (Fomicheva et al., 2020). (7) $P(\text{YES})$ is the probability of self-evaluation, as defined in Equation 1. (8) *Internal Confidence (w/ naive avg)* is a simplified variant of our proposed Internal Confidence. The difference is that we compute a naive average to aggregate all scores.

## D ADDITIONAL EXPERIMENTS

### D.1 CALIBRATION PERFORMANCE

Figure A1 illustrates the distributions of Internal Confidence for answerable versus non-answerable queries across three datasets—TriviaQA, SciQ, and GSM8K—using Phi-3.8B. In all cases, answerable queries (green) exhibit noticeably higher Internal Confidence, with distributions concentrated toward the upper end of the confidence range. In contrast, non-answerable queries (blue) show substantially lower Internal Confidence, typically forming broader or left-shifted distributions. This clear separation demonstrates that Internal Confidence effectively distinguishes between seen and unseen inputs, supporting its usefulness as an internal signal for assessing familiarity and reliability within the model.

| Method | ↑ AUROC | ↓ Time (s) | ↑ Speedup |
|---|---|---|---|
| **TriviaQA** | | | |
| Perplexity | 75.1 | 5.6 | 28× |
| Semantic Entropy | 72.3 | 139.5 | 698× |
| P(TRUE) | 65.2 | 22.5 | 113× |
| Lexical Similarity | 77.2 | 142.3 | 712× |
| SAR | 76.5 | 160.8 | 804× |
| MSP | 76.9 | 2.5 | 13× |
| CCP | 73.3 | 37.6 | 188× |
| EV Laplacian | **78.1** | 12.4 | 62× |
| Internal Confidence | 71.9 | **0.2** | — |
| **SciQ** | | | |
| Perplexity | **71.5** | 12.9 | 65× |
| Semantic Entropy | 66.3 | 132.8 | 664× |
| P(TRUE) | 60.4 | 22.1 | 111× |
| Lexical Similarity | 68.7 | 165.1 | 826× |
| SAR | 70.5 | 165.7 | 829× |
| MSP | 70.3 | 3.85 | 19× |
| CCP | 63.1 | 48.9 | 245× |
| EV Laplacian | 65.7 | 23.6 | 118× |
| Internal Confidence | 62.6 | **0.2** | — |

Table A1: Comparison of query-level Internal Confidence with answer-level uncertainty methods (Qwen-14B on TriviaQA and SciQ).

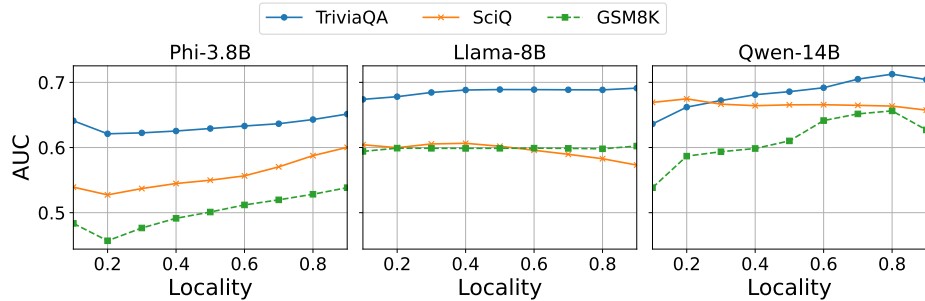

Figure A3: Impact of locality on validation set performance.

## D.2 INTERNAL CONFIDENCE DOES NOT RELY ON IN-CONTEXT LEARNING

Figure A2 shows the effect of the number of in-context learning example pairs ($k$-shot) on model performance across three datasets and models. Here, we randomly select $k$ pairs of positive and negative samples. We plot the AUROC as a function of $k$-shot values from 1 to 5. Overall, Llama-8B and Qwen-14B maintain relatively stable performance with slight improvements as $k$ increases, while Phi-3.8B exhibits more fluctuation, especially on TriviaQA. These results suggest that the benefit of additional in-context examples varies across both models and datasets. Therefore, our Internal Confidence can obtain strong performance even without in-context learning from examples, which can reduce the computational cost.

## D.3 IMPACT OF LOCALITY

Figure A3 presents the impact of locality on AUROC performance across three datasets (TriviaQA, SciQ, GSM8K) and three models (Phi-3.8B, Llama-8B, Qwen-14B). For Phi-3.8B, AUROC improves gradually with increasing locality across all datasets, with TriviaQA exhibiting consistently higher discriminability than SciQ and GSM8K. For Llama-8B, the performance remains fairly stable across different locality values, showing only minor fluctuations, particularly for SciQ and GSM8K. For Qwen-14B, the AUROC increases with the locality for all datasets up to a certain point, after which it either plateaus or slightly declines; this trend is most evident for GSM8K.

Locality has a non-trivial effect on the performance of Internal Confidence, and its optimal value varies slightly by model and dataset. Phi-3.8B and Qwen-14B benefit more clearly from tuning locality, while Llama-8B appears more robust to changes. Overall, high locality values often yield competitive or optimal performance.

## D.4 INTERNAL CONFIDENCE CAN BE GENERALIZED TO MORE CHALLENGING DATASETS

To validate whether our proposed internal confidence can be generalized to more challenging tasks, we conduct experiments on three additional datasets: (1) SimpleQA (Wei et al., 2024). This is a benchmark that evaluates the ability of language models to answer short, fact-relevant questions, which is less likely to be contaminated by the pre-training stage. (2) MuSiQue (Trivedi et al., 2022). This is a dataset that requires proper multihop reasoning, which is more difficult and harder to cheat via disconnected reasoning. (3) TruthfulQA (Lin et al., 2022). This is a benchmark to measure whether a language model is truthful in generating answers to questions. The authors crafted questions that some humans would answer falsely due to a false belief or misconception. To perform well, a model has to avoid generating false answers learned from imitating human texts. For each dataset, we use the validation or test partition for comparison, which contains a reasonable number of samples (2-4K). For the first two datasets, we apply the default configuration of internal confidence. Regarding the TruthfulQA dataset, we observe that the task exhibits a distinct decision center, which tends to appear in the middle layers rather than the upper layers across all three model architectures. For example, on a 100-sample held-out validation set, the decision centers appear at layers 9, 7, and 23 for Phi-3.8B, Llama-8B, and Qwen-14B, respectively. To consider this, we learn the decision center specifically for TruthfulQA using a 100-sample validation set. The overall

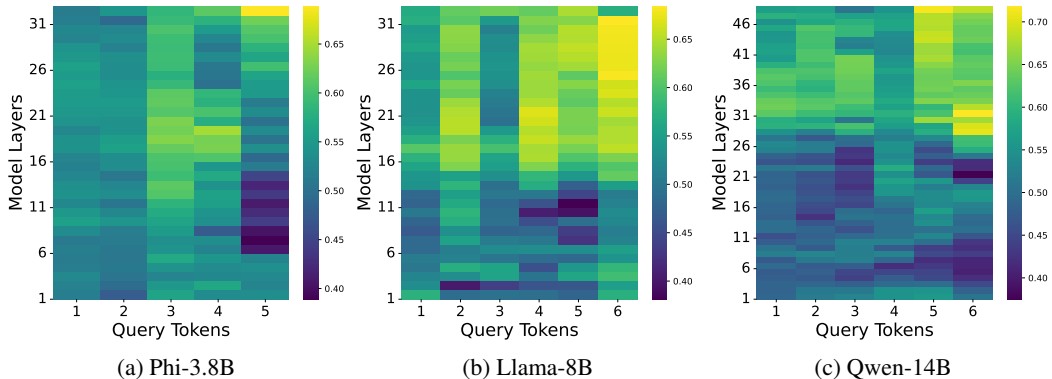

| | | |
|---|---|---|
| (a) Phi-3.8B | (b) Llama-8B | (c) Qwen-14B |

Figure A4: Learned decision centers of Phi-3.8B.

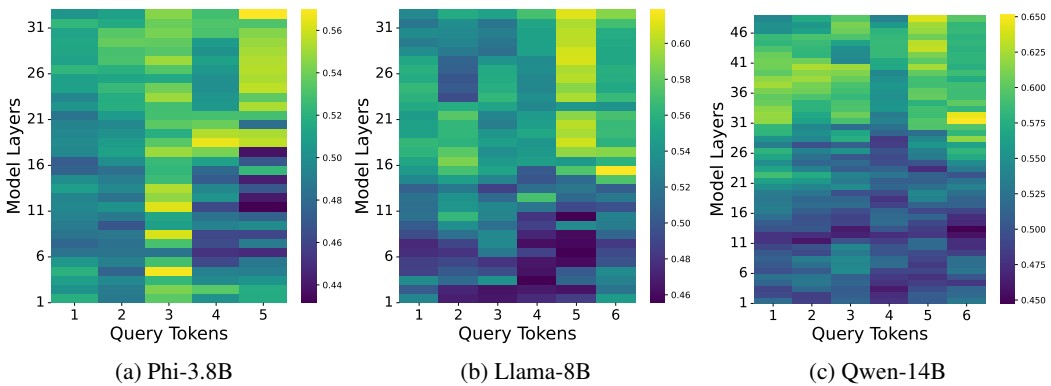

| | | |
|---|---|---|
| (a) Phi-3.8B | (b) Llama-8B | (c) Qwen-14B |

Figure A5: Learned decision centers of Llama-8B.

results are shown in Table A2. We can observe that our internal confidence can outperform other query-level uncertainty consistently across datasets and architectures.

## D.5 THE OPTIMAL DECISION CENTER VARIES ACROSS MODELS AND TASKS

We use the top right position as a default decision center, which offers a training-free and pragmatic solution, but it is the optimal center. We conduct experiments to study the learned decision center across different models and tasks. Figure A4, Figure A5, and Figure A6 show the locations of decision centers. We can observe that the center tends to appear at the top right place for TriviaQA and SciQ while the math reasoning task of GSM8K has a distinct behavior. The center is located in the lower layers. Although the current default center (top right) is sub-optimal, it offers a training-free, strong baseline, which can be generalized across different applications.

## E USE OF LARGE LANGUAGE MODELS

In this work, we employed LLMs in two complementary ways. First, LLMs were used to aid and polish the writing of the manuscript. This includes grammar checks and sentence polishing, mainly for readability and clarity. Second, LLMs were leveraged for retrieval, particularly in the section of related work. By querying LLMs to retrieve relevant references, we sought to identify additional references and obtain a comprehensive coverage of prior research.

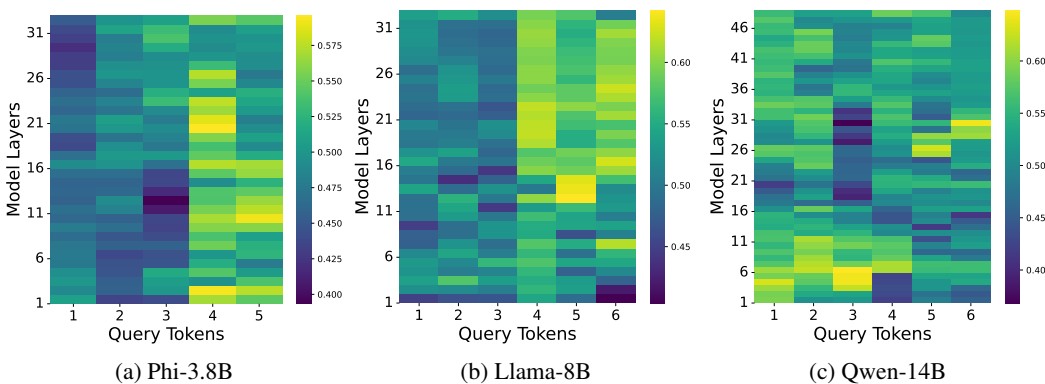

|  | (a) Phi-3.8B | (b) Llama-8B | (c) Qwen-14B |

Figure A6: Learned decision centers of Qwen-14B.

| Method | SimpleQA | | | MuSiQue | | | TruthfulQA | | | Avg | | |
|---|---|---|---|---|---|---|---|---|---|---|---|---|
|  | ↑ AUROC | ↑ PRR | ↓ ECE | ↑ AUROC | ↑ PRR | ↓ ECE | ↑ AUROC | ↑ PRR | ↓ ECE | ↑ AUROC | ↑ PRR | ↓ ECE |
| *Phi-3.8B* | | | | | | | | | | | | |
| Max($-\log p$) | 50.5 | 5.5 | —— | 53.2 | 4.7 | —— | 50.3 | -1.3 | —— | 51.3 | 3.0 | —— |
| Predictive Entropy | 54.0 | 5.9 | —— | **65.6** | 29.6 | —— | 55.7 | **15.6** | —— | 58.4 | 17.0 | —— |
| Min-K Entropy | 53.8 | 13.2 | —— | 59.9 | 21.9 | —— | 55.1 | 13.9 | —— | 56.3 | 16.3 | —— |
| Attentional Entropy | 50.1 | 5.1 | —— | 54.7 | 11.6 | —— | 52.4 | 8.3 | —— | 52.4 | 8.3 | —— |
| Perplexity | 52.4 | 5.8 | —— | 55.2 | 7.1 | —— | 54.1 | 3.8 | —— | 53.9 | 5.6 | —— |
| P(YES) (*top right*) | **61.3** | 17.8 | 18.8 | 65.4 | 29.8 | 9.5 | 39.6 | -16.2 | 27.1 | 55.4 | 10.5 | **18.5** |
| P(YES) (*naive avg*) | 59.8 | 17.8 | 69.9 | 65.5 | 29.5 | 63.2 | 49.3 | -2.0 | **25.5** | 58.2 | 15.1 | 52.9 |
| Internal Confidence | 61.2 | **26.1** | **18.2** | 65.5 | **30.2** | **9.3** | **56.4** | 13.2 | 40.7 | **61.0** | **23.2** | 22.7 |
| *Llama-8B* | | | | | | | | | | | | |
| Max($-\log p$) | 50.1 | -2.9 | —— | 53.2 | 6.3 | —— | 52.4 | 4.1 | —— | 51.9 | 2.5 | —— |
| Predictive Entropy | 49.1 | -0.9 | —— | 56.4 | 13.1 | —— | 60.0 | 13.7 | —— | 55.2 | 8.6 | —— |
| Min-K Entropy | 49.8 | 0.1 | —— | 56.0 | 15.2 | —— | 57.8 | 21.0 | —— | 54.5 | 12.1 | —— |
| Attentional Entropy | 48.6 | -4.2 | —— | 57.4 | 17.7 | —— | 53.3 | 9.6 | —— | 54.1 | 7.7 | —— |
| Perplexity | 50.1 | -3.8 | —— | 54.2 | 5.9 | —— | 54.3 | 4.8 | —— | 52.9 | 2.3 | —— |
| P(YES) (*top right*) | 53.6 | 5.2 | 78.9 | 64.1 | 27.3 | 74.5 | 43.3 | -11.9 | 55.7 | 53.7 | 6.9 | 69.7 |
| P(YES) (*naive avg*) | 54.9 | 5.8 | **28.5** | 63.2 | **31.9** | **18.5** | 47.0 | 1.9 | **3.4** | 55.0 | 13.2 | **16.8** |
| Internal Confidence | **55.6** | **11.6** | 67.4 | **64.3** | 29.8 | 74.4 | **63.2** | **26.8** | 15.4 | **61.0** | **22.7** | 52.4 |
| *Qwen-14B* | | | | | | | | | | | | |
| Max($-\log p$) | 50.8 | -1.2 | —— | 52.5 | 6.8 | —— | 51.0 | 4.7 | —— | 51.4 | 3.4 | —— |
| Predictive Entropy | 50.5 | 1.3 | —— | 53.8 | 9.4 | —— | **59.9** | 21.1 | —— | 54.7 | 10.6 | —— |
| Min-K Entropy | 51.8 | 8.1 | —— | 54.1 | 2.3 | —— | 58.1 | **22.8** | —— | 54.7 | 11.1 | —— |
| Attentional Entropy | 48.7 | -2.9 | —— | 54.9 | 13.2 | —— | 50.7 | 3.2 | —— | 51.4 | 4.5 | —— |
| Perplexity | 50.1 | -2.4 | —— | 53.7 | 10.7 | —— | 52.6 | 6.0 | —— | 52.1 | 4.8 | —— |
| P(YES) (*top right*) | 55.6 | 11.4 | 35.5 | 57.7 | 14.4 | 22.4 | 42.6 | -19.0 | 53.4 | 52.0 | 2.3 | 37.1 |
| P(YES) (*naive avg*) | **57.6** | **16.6** | **4.0** | **58.4** | **16.9** | 6.5 | 49.7 | -3.5 | 8.6 | 55.2 | 10.0 | 6.4 |
| Internal Confidence | 56.0 | 13.2 | 10.3 | 57.6 | 14.6 | **5.4** | 58.0 | 16.4 | **0.5** | **57.2** | **14.7** | **5.4** |

Table A2: Additional results of different query-level uncertainty estimation methods.

