# OpenReview forum: "Query-Level Uncertainty in Large Language Models"
_ICLR.cc/2026/Conference — ICLR 2026 Poster_

### Official Review · Reviewer_pkEN · 2025-10-25

**Soundness:** 2
**Presentation:** 1
**Contribution:** 3
**Rating:** 4
**Confidence:** 3

**Summary:**

This paper proposes a training-free, pre-generation method Internal Confidence to investigate the knowledge boundaries of LLMs and reduce inference costs.
The authors prompt LLMs with a yes–no question to self-assess whether they can answer a given query before generating an answer, and they adopt layer-level consistency to measure the reliability of the answers.
Experiments show some efficiency improvements over simple baselines.

**Strengths:**

The idea of identifying an LLM’s knowledge boundaries before allowing it to generate answers or take agentic actions is both interesting and impactful.
As the paper demonstrates, when used properly, such methods can significantly reduce the costs of applying LLMs to downstream tasks.

**Weaknesses:**

- The method description is confusing, with some simple concepts made complex and other critical details missing.
- The experimental results are not sufficiently robust.
- The article is ambiguous throughout.
- Although not required, the authors are encouraged to discuss the limitations of their work.

**Questions:**

- "We define a query as being within the model’s knowledge boundary if the LLM can produce a correct answer under greedy decoding, i.e., by selecting the highest-probability token at each step without sampling." --- I’m not sure how or if the answers’ correctness correlates with the model's knowledge boundary. Are there any previous works that support this statement?

- Why introduce the unmapping weights instead of directly using the last-layer pre-activation logits for a clearer description? Did you modify the unmapping weights to keep only the rows corresponding to Yes and No tokens for computing the logits?

- Why introduce Lines 211--235? Are the parameters theta retrained or from external layers? Why not simply state "pre-/post-activation output of each Transformer block"?

- It is unclear what the word "query" refers to in different contexts. In general, it seems to indicate the model's prompt; but in the example in Lines 194–195, it seems to refer to the question; and in the discussion in Lines 237–245, both meanings do not make sense, and it seems "query" here refers to both the prompt and the model's answer. This is extremely confusing.

- In Figure 3 and Lines 237--245, it is unclear which tokens these probabilities/scores are associated with. Suppose these are answer tokens; then we may expect these results, since the last token is highly likely to be EOS, so the next-to-last token is the actual Yes/No token. If not, I’m not sure how to interpret these results. In either case, the discussion does not carry much useful information.

- Lines 253--258: I don't get the value of this paragraph. Equation (3) looks like a simple weighted average, so why form it into a "hierarchical aggregation" process?

- Equations (4) and (5): How is delta related to epsilon? How is locality related to the weights w?

- When calculating the runtime, is the time for generating the actual answers included? Why not include predictive entropy in Table 2? Why not include length-normalized predictive entropy in Tables 1 and 2? Semantic aggregation methods such as Semantic Entropy and SAR are not designed for a task with a deterministic answer, such as GSM8k; why include them in Table 2? Why not use TriviaQA?

- Line 376: Area under what curve? I know the answer, but it may confuse others who are not so familiar with UQ.

- Line 315: What is the necessity of evaluating the reasoning steps for GSM8k? What would happen if we do not do so?

---

> ### Author Response · Authors · 2025-11-22
> **response (1/4)**
>
> Dear Reviewer pkEN,
>
> We thank you for your detailed comments and useful feedback. We appreciate the time you invested in reviewing our paper. We are encouraged that you find our idea interesting and impactful, and our method can significantly reduce the cost of LLM inference. We have revised our manuscript according to your feedback.
>
> **Q1: We define a query as being within the model’s knowledge boundary if the LLM can produce a correct answer under greedy decoding**
>
> > by selecting the highest-probability token at each step without sampling." --- I’m not sure how or if the answers’ correctness correlates with the model's knowledge boundary. Are there any previous works that support this statement?
>
> **Answer:**
>
> We have revised the definition in Section 3.1 to clarify this. Below is the concrete answers.
>
> Our definition of the knowledge boundary is aligned with the concept of epistemic uncertainty (we have described this in the original submission in Section A.1), which stems from a lack of knowledge, often due to insufficient training data and limited model capacity [1, 2]. The survey paper also introduces a similar concept, Parametric Knowledge Boundary, which defines the abstract knowledge boundary for a specific LLM. The parametric knowledge boundary of an LLM is the set of all knowledge encoded in its parameters that can be verified by at least one input–output pair.
>
> In the original submission, we emphasized that the setting of greedy decoding serves as a heuristic indicator of internal knowledge, rather than an absolute measure. *"While greedy decoding ensures deterministic measurement, it may not always reflect the optimal performance of a model~\citep{song2024good}, as alternative decoding strategies like beam search may elicit a better answer.
> Therefore, this pragmatic framework serves as a heuristic indicator of internal knowledge, rather than an absolute measure.
> We use this standard to evaluate the estimated query-level uncertainty, i.e., a lower uncertainty indicates a model is more likely to output the correct answer. "*
>
> To clarify this, we added further description in this section (blue texts) to clarify why we stick with greedy decoding:
> * Efficiency. Our method treats a successful greedy decode as a signal that the model knows how to answer, which is a fast proxy. In contrast, non-greedy decoding requires the configuration of beam sizes, probability thresholds, and sampling numbers, which complicates both the definition of the knowledge boundary and the assessment cost. How to define a correct answer in a non-greedy decoding setting is tricky.
> * Reproducibility. Greedy decoding outputs a single deterministic output for a given input, which offers a stable and reproducible baseline and benchmark.
>
> At the same time, we have added experiments to study the query-level uncertainty under a non-greedy decoding setting.
> Specifically, we enable sampling, set the beam size to 5, and use a temperature of 1.0. We assess different query-level uncertainty methods on Llama-8B using the TriviaQA dataset.
> For each query, we prompt the model to generate 10 answers. If the model produces the correct answer in more than 5 out of the 10 attempts, we consider the query to fall within the model’s knowledge boundary.
> The results are shown in the Table below. We can see that our Internal Confidence can identify known and unknown queries better than other competitors, and the scores (also the ranking of each method) are highly similar to the results under greedy decoding. These consistent findings confirm that our greedy decoding setting is a simple yet effective and robust way to measure the knowledge boundary.
>
> | Method                 | AUC ↑ | PRR ↑ | ECE ↓ |
> |------------------------|-------|-------|-------|
> | Max(– log p)           | 54.4  | 11.6  | ---   |
> | Predictive Entropy     | 57.6  | 10.8  | ---   |
> | Min-K Entropy          | 59.1  | 19.1  | ---   |
> | Attentional Entropy    | 61.2  | 23.5  | ---   |
> | Perplexity             | 61.0  | 24.8  | ---   |
> | P(YES) (top right)     | 57.1  | 14.0  | 26.0  |
> | P(YES) (naive avg)     | 66.3  | 30.0  | **18.1** |
> | Internal Confidence    | **71.7** | **45.7** | 19.9 |
>
>
> [1] Aleatory and epistemic uncertainty in probability elicitation with an example from hazardous waste management.
>
> [2] Aleatory or epistemic? Does it Matter?
>
> [3] Knowledge Boundary of Large Language Models: A Survey

---

> ### Author Response · Authors · 2025-11-22
> **response (2/4)**
>
> **Q2: Why introduce the unmapping weights instead of directly using the last-layer pre-activation logits for a clearer description?**
>
> > Did you modify the unmapping weights to keep only the rows corresponding to Yes and No tokens for computing the logits?
>
> **Answer:**
>
> We introduce the unmapping weight to have a meaningful confidence score (the probability), and we do not modify it.
>
> Our internal confidence performs an aggregation across different layers and tokens. In the original Transformer paper, the unmapping matrix is applied only to the last-layer hidden states $h_{N}^{(L)}$ for computing the next token probability.  We introduce the unembedding to explain how to apply it to intermediate layers to obtain internal P(Yes). The logits are not directly used since we wanted to have a human-readable and comparable probability, which can be used to rank the confidence scores in downstream tasks.  However, the raw logits are not directly interpretable in this way. We have clarified this in line 211.
>
> **Q3: Why introduce Lines 211--235? Are the parameters theta retrained or from external layers? Why not simply state "pre-/post-activation output of each Transformer block"?**
>
> **Answer:**
>
> In a transformer block, the pre-activation means the value before applying the nonlinearity $\phi$, while the post-activation means the value after applying $\phi$. In our method, the hidden states $h$ are used rather than the activation-level information, which requires a residual add and a linear transformation of the post-activation. This hidden state $h$ can be projected to the vocabulary space for obtaining the probability distribution. After, we use lines 211-235  and equations to describe how to obtain $N \times L$ latent representations, which is important to introduce the next aggregation step. We hope that our content in the manuscript is precise and rigorous.
>
>
> **Q4: It is unclear what the word "query" refers to in different contexts.**
>
> > In general, it seems to indicate the model's prompt; but in the example in Lines 194–195, it seems to refer to the question; and in the discussion in Lines 237–245, both meanings do not make sense, and it seems "query" here refers to both the prompt and the model's answer. This is extremely confusing.
>
> **Answer:**
>
> In the original submission, we have provided the definition of the query (the beginning of Section 3.1).
> "Given a query (including prompt tokens) $\mathbf{x} = (x_1, \ldots, x_N)$,
> we aim to quantify the query-level uncertainty, $U(\mathbf{x})$, without generating an answer $\mathbf{y}$. This differs from existing uncertainty approaches that estimate the uncertainty associated with a specific generated answer, an answer-level uncertainty that can be denoted as $U(\mathbf{x},\mathbf{y})$."
>
> Therefore, the query includes both the question itself and prompt tokens, while the model's answer is completely not involved in our approach.
>
> **Q5: In Figure 3 and Lines 237--245, it is unclear which tokens these probabilities/scores are associated with.**
>
> > Suppose these are answer tokens; then we may expect these results, since the last token is highly likely to be EOS, so the next-to-last token is the actual Yes/No token. If not, I’m not sure how to interpret these results. In either case, the discussion does not carry much useful information.
>
> **Answer:**
>
> The central research question in this work is: **Given a query, can we determine whether the model can address it before generating any tokens?** To this end, we propose query-level uncertainty, which is able to identify the knowledge boundary of a model at the pre-generation stage. Therefore, these probabilities are associated with each token and layer within the query.  Our method does not require generating answer tokens.
>
> **Q6: Lines 253--258: I don't get the value of this paragraph. Equation (3) looks like a simple weighted average, so why form it into a "hierarchical aggregation" process?**
>
> **Answer:**
>
> We agree that, mathematically, Equation (3) is a weighted sum. Our intention in describing it as a hierarchical aggregation was not to claim additional mathematical complexity, but to make clear the structure of the weighting.
> This aggregation contains two steps:
> * Across layers: for each token, we first aggregate confidence scores over layers using $w^{layer}$, corresponding to an ensemble over different representation depths.
> * Across tokens: we then aggregate these token-level results using $w^{token}$, which controls how different positions in the sequence contribute to the final confidence score.
> The hierarchical view clarifies how to perform this aggregation in a structured way.

---

> ### Author Response · Authors · 2025-11-22
> **response (3/4)**
>
> **Q7: Equations (4) and (5): How is delta related to epsilon? How is locality related to the weights w?**
>
> **Answer:**
>
> We have revised our manuscript, and now we use a unified delta to represent the weight (see equation 5 and the surrounding context).
>
> Locality ($\in [0, 1]$) is a metric that measures the extent to which weights are concentrated in adjacent positions of a center.
> A value of 1 implies that the vector perfectly satisfies the locality property, which means weights are extremely concentrated at the decision center. A low locality means weights are more uniformly assigned to neighborhoods.
> For example, given a sequence of length 5 and a weight vector for the first position `[1, 0, 0, 0, 0]`, the locality is **1**,  which means it perfectly satisfies the locality property.  In contrast, the locality is **1/16** if the weight attends only the last position, as in `[0, 0, 0, 0, 1]`.
>
> Figure 3c  plots the weights obtained from Equation 4 for varying degrees of locality.
> This shows that we can account for the influence of neighboring layers and tokens during the averaging process.
>
>
> **Q8: When calculating the runtime, is the time for generating the actual answers included?**
>
> > Why not include predictive entropy in Table 2? Why not include length-normalized predictive entropy in Tables 1 and 2? Semantic aggregation methods such as Semantic Entropy and SAR are not designed for a task with a deterministic answer, such as GSM8k; why include them in Table 2? Why not use TriviaQA?
>
> **Answer:**
>
> In this work, we proposed the query-level uncertainty, which can identify if a model can distinguish known and unknown queries without generating any answer tokens (at the pre-generation stage). In contrast, existing approaches, **answer-level uncertainty** require the model's answer to estimate uncertainty. Our approach is completely different from existing work, and the main benefit of using our method is that it can have very competitive performance while being extremely faster.
>
> We compare query-level and answer-level approaches in a separate way, which would be clearer and fairer.  Table 1 shows the **overall results of different query-level uncertainty estimation methods.** while Table 2 shows **comparison with answer-level uncertainty methods**.
>
> We have added stronger baselines in Table 2 and Table A1, which compare our internal confidence (*query-level*) to other answer-level approaches across three datasets.  The table below shows a comprehensive comparison, which supports that our method can have competitive performance while being extremely faster.
>
> | Method               | TriviaQA AUC | TriviaQA Time | TriviaQA Speedup | SciQ AUC | SciQ Time | SciQ Speedup | GSM8K AUC | GSM8K Time | GSM8K Speedup |
> |----------------------|--------------|----------------|-------------------|----------|-----------|---------------|-----------|------------|----------------|
> | CCP                  | 73.3         | 37.6           | 188X              | 63.1     | 48.9      | 245X          | 64.2      | 61.7       | 206X           |
> | MSP                  | 76.9         | 2.49           | 13X               | **70.3** | 3.85      | 19X           | **68.5**  | 25.1       | 84X            |
> | EigValLaplacian      | **78.1**     | 12.4           | 62X               | 65.7     | 23.6      | 118X          | 56.7      | 153.8      | 512X           |
> | Lexical Similarity   | 77.2         | 140.3          | 702X              | 68.7     | 165.7     | 829X          | 62.4      | 22.3       | 74X            |
> | DegMat               | 75.0         | 12.2           | 61X               | 63.1     | 25.1      | 125X          | 62.1      | 153.9      | 513X           |
> | TokenSAR             | 78.0         | 17.4           | 87X               | 50.1     | 27.1      | 135X          | 62.2      | 189.5      | 632X           |
> | Internal Confidence  | 71.9         | **0.2**        | —                 | 62.6     | **0.2**   | —             | 66.8      | **0.3**    | —              |

---

> ### Author Response · Authors · 2025-11-22
> **response (4/4)**
>
> **Q9: Line 376: Area under what curve? I know the answer, but it may confuse others who are not so familiar with UQ.**
>
> **Answer:**
>
> We revised this description (see line 391). Now it is "we adopt the Area Under the Receiver Operating Characteristic Curve (AUC)"
>
> **Q10: Line 315: What is the necessity of evaluating the reasoning steps for GSM8k? What would happen if we do not do so?**
>
> **Answer:**
>
> We evaluate the reasoning steps on GSM8K because verifying the reasoning chain is essential to ensure the model truly understands the problem rather than outputting the correct results by chance. We have added this in line 377
>
>
> **Q11: Although not required, the authors are encouraged to discuss the limitations of their work.**
>
> **Answer:**
>
> We have added the limitation section after the conclusion.

---

> > ### Comment · Reviewer_pkEN · 2025-11-22
> >
> > Q1 - I do not have access to reference 1, and did not find evidence supporting "define a query as being within the model’s knowledge boundary if the LLM can produce a correct answer under greedy decoding" in references 2 and 3. Do you mind pointing out where in the papers your claim is supported?
> >
> > Q2&3 I respect your choice but I still find it unnecessary to include these discussion since they are common knowledge to the (competent) ICLR community. A simple sentence "h^(l) is the output of the l-th Transformer block", IMO, is good enough to cover all of these content. BTW, pre-/post-activation is specific to the last layer, targeting "the unmapping layer", which is more frequently referred to as "unembedding" actually.
> >
> > Q4 The concept of "query" still looks confusing and inconsistent. Including some examples might help.
> >
> > Q5 I'm not sure what you are talking about but it does not seem to be an answer to the question. This goes back to Q4.
> >
> > Q6 Again, this should be common knowledge that do not worth a discussion.
> >
> > Q7 If its a typo, just say its a typo. I do understand math.
> >
> > Q8 But the ultimate goal is still to get answers from LLMs, right? Suppose now you know the epistemic uncertainty is low and you do the actual inference (whose time is not added to table 1 I guess?), you are still unable to know whether your answer is accurate. This leads to the question of how you calculated your ECE, which is not specified either. Besides, some questions are ignored.
> >
> > Q10 Then why don't you do the same to other datasets. You can always ask the model to give a reasoning chain.

---

> > > ### Comment · Reviewer_pkEN · 2025-11-22
> > >
> > > BTW, providing the code that reproduces your results would be more useful than a toy example.

---

> ### Author Response · Authors · 2025-12-01
>
> Dear Reviewer pkEN,
>
> We would like to thank you for your questions and feedback. We appreciate your engaged discussions.
>
> **Q1 - I do not have access to reference 1, and did not find evidence supporting "define a query as being within the model’s knowledge boundary if the LLM can produce a correct answer under greedy decoding" in references 2 and 3. Do you mind pointing out where in the papers your claim is supported?**
>
> **Answer:**
>
> Our definition of the knowledge boundary is aligned with the parametric knowledge boundary [1] (in Section 2 of the original paper ). The boundary of a model is the set of all knowledge encoded in its parameters that can be verified by at least one input–output pair.
>
> [1] Li, Moxin, et al. "Knowledge boundary of large language models: A survey." Proceedings of the 63rd Annual Meeting of the Association for Computational Linguistics (Volume 1: Long Papers). 2025.
>
>
> **Q2:Q2&3 I respect your choice but I still find it unnecessary to include these discussion since they are common knowledge to the (competent) ICLR community. A simple sentence "h^(l) is the output of the l-th Transformer block", IMO, is good enough to cover all of these content. BTW, pre-/post-activation is specific to the last layer, targeting "the unmapping layer", which is more frequently referred to as "unembedding" actually.**
>
> **Answer:**
>
> We assure the reviewer that our intention is not to complicate the manuscript with unnecessary notations.
> On the contrary, we have deliberately used simple and minimal notation to keep the presentation precise and easy to follow.
>
> Regarding your question, the activation and hidden states are not strictly equivalent, and we introduce the notation of the hidden state to clarify this. Moreover, we apply the unembeeding to each intermediate layer to compute the internal confidence, which is a key step in our method and is not a standard part of the original transformer. Without these notations, several key technical details would be lost or remain ambiguous.
>
> **Q4 The concept of "query" still looks confusing and inconsistent. Including some examples might help.**
>
> **Answer:**
>
> Thanks for your suggestion. We have added the prompt used in our experiment (see line 830). We hope this update can help you understand the concept of query well in our manuscript.
>
> **Q7 If its a typo, just say its a typo. I do understand math**
>
> **Answer:**
>
> We greatly respect your expertise. While there is indeed no typo in the original equation, we agree that its form is not fully consistent. To improve clarity and readability, we have adopted a unified notation $\delta$, which we hope will make the presentation easier to follow.
>
> **Q8 But the ultimate goal is still to get answers from LLMs, right? Suppose now you know the epistemic uncertainty is low and you do the actual inference (whose time is not added to table 1 I guess?), you are still unable to know whether your answer is accurate. This leads to the question of how you calculated your ECE, which is not specified either. Besides, some questions are ignored.**
>
> **Answer:**
>
> Thanks for your question.
>
> The ultimate goal is to estimate the epistemic uncertainty without generating answers, which is the core idea of this work (the problem definition is clearly stated in Section 3.1).
> In Table 1, all baselines we compare against are also query-level uncertainty methods, meaning they do not require performing “actual inference” to obtain an answer, and thus they have similar running time. However, to evaluate the performance of these methods, we must still refer to the correct answers.
>
> **Q10 Then why don't you do the same to other datasets. You can always ask the model to give a reasoning chain.**
>
> **Answer:**
>
> Thanks for your interesting question.
>
> The experiments were conducted in accordance with prior work [1]. We use a reasoning chain for GSM8K because it is a multi-step mathematical reasoning task, where intermediate steps are important for the model to arrive at a correct solution. It is possible that a model gives the correct answer with a wrong reasoning chain. In contrast, TriviaQA and SciQ are short-form factual QA. For these datasets, a direct answer is sufficient for the evaluation step.
>
> [1] Kadavath, Saurav, Tom Conerly, Amanda Askell, Tom Henighan, Dawn Drain, Ethan Perez, Nicholas Schiefer et al. "Language models (mostly) know what they know." arXiv preprint arXiv:2207.05221 (2022).
>
> **Q: BTW, providing the code that reproduces your results would be more useful than a toy example.**
>
> **Answer:**
>
> Thank you for your question. We have uploaded the key implementation details for Internal Confidence as well as the baseline methods, which would help clarify the specific prompts and model configurations used in our experiments. We strongly support open science and will release a public repository for reproducibility.

---

### Official Review · Reviewer_hjve · 2025-10-28

**Soundness:** 3
**Presentation:** 3
**Contribution:** 2
**Rating:** 4
**Confidence:** 3

**Summary:**

This paper introduces Query-Level Uncertainty, which estimates whether an LLM can answer a query before generating tokens, addressing efficiency and trustworthiness concerns. The proposed Internal Confidence method aggregates P(YES) signals across layers and tokens using attenuated encoding weights, obtaining calibrated query-level uncertainty in a single forward pass. Experiments on factual QA and mathematical reasoning tasks show improvements over adapted answer-level methods. Applications to efficient RAG and model cascading demonstrate practical benefits with 30-600× speedups over answer-level approaches.

**Strengths:**

Query-level uncertainty is a genuinely different and practical perspective from existing answer-level methods. Rather than generating long answers to assess uncertainty, the method predicts answerability before token generation, directly addressing efficiency bottlenecks in real-world systems. The Internal Confidence method is elegantly simple—it aggregates P(YES) signals across layers and tokens with attenuated encoding weights, producing calibrated uncertainty in a single forward pass without requiring training.

The empirical results are strong and consistent. The method shows improvements over adapted baselines across three diverse datasets and works reliably across different model sizes (Phi-3.8B, Llama-8B, Qwen-14B). More impressively, it achieves 30-600× speedups compared to answer-level approaches like Semantic Entropy and SAR, making it genuinely deployable. Ablation studies systematically examine the impact of locality and hyperparameters, providing confidence in design choices.

The experimental design is rigorous with proper out-of-domain evaluation demonstrating robustness to domain shift. The applications to efficient RAG and model cascading are immediately actionable, with identified "optimal points" that practitioners can use directly. The paper is well-written with clear motivation and effective visualizations that make the approach accessible.

**Weaknesses:**

The theoretical justification is weak. While P(YES) seems intuitive as a proxy for query answerability, the connection to actual answering capability is assumed rather than proven. Similarly, attenuated encoding is presented as effective, but other aggregation schemes aren't systematically compared to justify this choice. The "decision center" concept, which is central to the method, lacks theoretical grounding—the paper doesn't explain why the top-right position is optimal or how it should adapt across different models and tasks.

The evaluation scope is narrow and raises generalization concerns. Experiments are limited to factual QA and mathematical reasoning across just three datasets with 10K samples each. There's no evaluation on open-ended tasks, creative writing, code generation, or other generation types. Additionally, the method assumes greedy decoding for determining ground truth, which may not reflect real model behavior under beam search or sampling-based decoding strategies.

Ground truth definition is somewhat arbitrary and validation is incomplete. Correctness is defined solely by greedy decoding accuracy, and the paper lacks comparison against human-annotated "answerability" judgments. It remains unclear whether the model's internal uncertainty truly reflects knowledge or is confounded by input formatting, instruction phrasing, or other factors. The paper also doesn't analyze systematic biases—does the method correlate with question difficulty, answer length, or other confounds?

The method's claims of being "training-free" are overstated. While no training is required, hyperparameter selection (α value and decision center location) depends on validation data, and cross-dataset transfer shows these vary by task and model. The paper uses fixed hyperparameters across all settings despite acknowledging they're suboptimal, and the inconsistency between task-specific optimal centers and the fixed top-right choice isn't well justified.

Baseline comparisons are limited to adaptations of answer-level methods, with no comparison to other query-level uncertainty approaches or simpler heuristics like maximum token probability of the query. Code availability isn't mentioned, and specific prompts and model configurations lack complete documentation, hampering reproducibility and follow-up work.

**Questions:**

1. Why is P(YES) specifically the right formulation? Have you tried other self-evaluations (e.g., confidence level on a 1-10 scale)?

2. How does the method perform on adversarial inputs where models give confident wrong answers?

3. Can you provide correlation analysis between Internal Confidence scores and actual greedy decoding accuracy?

4. How sensitive are results to the specific threshold used for deciding "know/don't know"?

5. For the decision center analysis (Figure 3b), why does performance peak at h_5^(27) rather than final layer?

6. Have you tested on models with different architectures (MoE, retrieval-augmented)?

---

> ### Author Response · Authors · 2025-11-22
> **response (1/5)**
>
> Dear Reviewer hjve,
>
> We would like to thank you for your critical comments and insightful suggestions. We truly appreciate the time and effort you devoted to reviewing our paper. We are encouraged that you find our proposed query-level uncertainty measure both genuinely novel and practically useful, and our empirical results are strong and consistent. The manuscript has been revised according to your feedback.
>
> **Q1: The theoretical justification is weak. While P(YES) seems intuitive as a proxy for query answerability, the connection to actual answering capability is assumed rather than proven.**
>
> **Answer:**
>
> While P(YES) is not theoretically guaranteed, we consider it a pragmatically useful proxy and an efficient heuristic.
> According to prior studies [1,2], LLMs can self-evaluate whether they know the answer to a question without reference to any specific proposed answer. At the same time, a recent work indicates that answerable and unanswerable questions are also linearly separable in hidden states [3]. Therefore, our Internal Confidence can be regarded as a training-free linear probe for answerability, which captures the model’s implicit internal boundary between queries the model believes it can answer and those it believes it cannot. We have clarified this at the beginning of Section 3.2.
>
> [1] Kadavath et al. Language models (mostly) know what they know
>
> [2] Li et al. Knowledge Boundary of Large Language Models: A Survey
>
> [3] Slobodkin et al. The Curious Case of Hallucinatory (Un)answerability: Finding Truths in the Hidden States of Over-Confident Large Language Models
>
> **Q2: attenuated encoding is presented as effective, but other aggregation schemes aren't systematically compared to justify this choice.**
>
> **Answer:**
>
> We have compared our attenuated encoding to a naive weighted average (*P(YES) (naive avg)*). Table 1 shows that our encoding can outperform a simple aggregation strategy across models and tasks. Additionally, we have tested different variants of this encoding. Figure A3 presents the impact of locality on AUC performance across three datasets. The locality controls how to perform this aggregation.
>
> **Q3: The "decision center" concept, which is central to the method, lacks theoretical grounding—the paper doesn't explain why the top-right position is optimal or how it should adapt across different models and tasks.**
>
> **Answer:**
>
> In the initial submission, we stressed that *"In our implementation, we adopt the top-right cell (corresponding to the last token and last layer) as the decision center, since we observe that the decision center tends to be located near the later layers and final tokens across various architectures and tasks. While, in principle, the optimal decision center may also lie elsewhere, identifying such an optimal center would require a hold-out set of training data, which conflicts with our goal of developing a training-free approach."*
> We therefore do not claim the top-right position is optimal, but it serves a practical implementation that works well empirically.
>
> We conducted experiments to study the learned decision center across different models and tasks. Figures A4, A5, and A6 in the appendix show the locations of decision centers. We observe that the center tends to appear at the top right place for TriviaQA and SciQ, while the math reasoning task of GSM8K has a distinct behavior. The center is located in the lower layers. Although the current default center (top right) is sub-optimal, it offers a training-free, strong baseline, which can be generalized across different applications. We have added Section C.5 in the manuscript.

---

> ### Author Response · Authors · 2025-11-22
> **response (2/5)**
>
> **Q4: The evaluation scope is narrow and raises generalization concerns**
>
> > Experiments are limited to factual QA and mathematical reasoning across just three datasets with 10K samples each. There's no evaluation on open-ended tasks, creative writing, code generation, or other generation types.
>
> **Answer:**
>
> We conducted additional experiments to include more challenging datasets: SimpleQA, TruthfulQA, and MuSiQue, which aim to cover less contaminated, open-ended, and multi-hop reasoning benchmarks. The details and results are shown in Section C.4 in the manuscript. We also show the results of Phi below due to the space limit. We can observe that our Internal Confidence can still outperform other query-level baselines on more challenging datasets, which demonstrates the generality of our proposed method.
>
> | Model         | Method                       | SimpleQA AUC | PRR | ECE | MuSiQue AUC | PRR | ECE | TruthfulQA AUC | TruthfulQA PRR | TruthfulQA ECE | Avg AUC | Avg PRR | Avg ECE |
> |---------------|------------------------------|-------------:|-------------:|-------------:|-------------:|-------------:|-------------:|---------------:|----------------|---------------:|--------:|--------:|--------:|
> | **Phi-3.8B**  | Max (-log p)                 | 50.5        | 5.5          | –            | 53.2         | 4.7          | –            | 50.3           | –1.3            | –             | 51.3    | 3.0     | –      |
> |               | Predictive Entropy           | 54.0        | 5.9          | –            | **65.6**      | 29.6         | –            | 55.7           | **15.6**         | –             | 58.4    | 17.0    | –      |
> |               | Min-K Entropy                | 53.8        | 13.2         | –            | 59.9         | 21.9         | –            | 55.1           | 13.9            | –             | 56.3    | 16.3    | –      |
> |               | Attentional Entropy          | 50.1        | 5.1          | –            | 54.7         | 11.6         | –            | 52.4           | 8.3             | –             | 52.4    | 8.3     | –      |
> |               | Perplexity                   | 52.4        | 5.8          | –            | 55.2         | 7.1          | –            | 54.1           | 3.8             | –             | 53.9    | 5.6     | –      |
> |               | P(Yes)  (top right)         | **61.3**     | 17.8         | 18.8         | 65.4         | 29.8         | 9.5          | 39.6           | –16.2           | 27.1          | 55.4    | 10.5    | **18.5**|
> |               | P(Yes)  (naive avg)         | 59.8        | 17.8         | 69.9         | 65.5         | 29.5         | 63.2         | 49.3           | –2.0            | **25.5**       | 58.2    | 15.1    | 52.9   |
> |               | Internal Confidence          | 61.2        | **26.1**     | **18.2**     | 65.5         | **30.2**     | **9.3**       | **56.4**        | 13.2            | 40.7          | **61.0** | **23.2**| 22.7   |

---

> ### Author Response · Authors · 2025-11-22
> **response (3/5)**
>
> **Q5: Ground truth definition is somewhat arbitrary and validation is incomplete.**
>
> > Additionally, the method assumes greedy decoding for determining ground truth, which may not reflect real model behavior under beam search or sampling-based decoding strategies. Correctness is defined solely by greedy decoding accuracy, and the paper lacks comparison against human-annotated "answerability" judgments. It remains unclear whether the model's internal uncertainty truly reflects knowledge or is confounded by input formatting, instruction phrasing, or other factors. The paper also doesn't analyze systematic biases—does the method correlate with question difficulty, answer length, or other confounds?
>
> **Answer:**
>
> In the original submission, we emphasized that the setting of greedy decoding serves as a heuristic indicator of internal knowledge, rather than an absolute measure. *"While greedy decoding ensures deterministic measurement, it may not always reflect the optimal performance of a model~\citep{song2024good}, as alternative decoding strategies like beam search may elicit a better answer.
> Therefore, this pragmatic framework serves as a heuristic indicator of internal knowledge, rather than an absolute measure.
> We use this standard to evaluate the estimated query-level uncertainty, i.e., a lower uncertainty indicates a model is more likely to output the correct answer. "*
>
> To clarify this, we added further description in this section (blue texts) to clarify why we stick with greedy decoding:
> * Efficiency. Our method treats a successful greedy decode as a signal that the model knows how to answer, which is a fast proxy. In contrast, non-greedy decoding requires the configuration of beam sizes, probability thresholds, and sampling numbers, which complicates both the definition of the knowledge boundary and the assessment cost. How to define a correct answer in a non-greedy decoding setting is tricky.
> * Reproducibility. Greedy decoding outputs a single deterministic output for a given input, which offers a stable and reproducible baseline and benchmark.
>
> At the same time, we have added experiments to study the query-level uncertainty under a non-greedy decoding setting.
> Specifically, we enable sampling, set the beam size to 5, and use a temperature of 1.0. We assess different query-level uncertainty methods on Llama-8B using the TriviaQA dataset.
> For each query, we prompt the model to generate 10 answers. If the model produces the correct answer in more than 5 out of the 10 attempts, we consider the query to fall within the model’s knowledge boundary.
> The results are shown in the Table below. We can see that our Internal Confidence can identify known and unknown queries better than other competitors, and the scores (also the ranking of each method) are highly similar to the results under greedy decoding. These consistent findings confirm that our greedy decoding setting is a simple yet effective and robust way to measure the knowledge boundary.
>
>
> | Method                 | AUC ↑ | PRR ↑ | ECE ↓ |
> |------------------------|-------|-------|-------|
> | Max(– log p)           | 54.4  | 11.6  | ---   |
> | Predictive Entropy     | 57.6  | 10.8  | ---   |
> | Min-K Entropy          | 59.1  | 19.1  | ---   |
> | Attentional Entropy    | 61.2  | 23.5  | ---   |
> | Perplexity             | 61.0  | 24.8  | ---   |
> | P(YES) (top right)     | 57.1  | 14.0  | 26.0  |
> | P(YES) (naive avg)     | 66.3  | 30.0  | **18.1** |
> | Internal Confidence    | **71.7** | **45.7** | 19.9 |
>
> **Q6: The method's claims of being "training-free" are overstated.**
>
> > While no training is required, hyperparameter selection (α value and decision center location) depends on validation data, and cross-dataset transfer shows these vary by task and model. The paper uses fixed hyperparameters across all settings despite acknowledging they're suboptimal, and the inconsistency between task-specific optimal centers and the fixed top-right choice isn't well justified.
>
> **Answer:**
>
> Our initial motivation is to provide a simple and strong baseline for knowledge boundary detection, which can be generalized to unseen datasets and tasks.  We have applied our default Internal Confidence to more complex datasets, which do not provide validation sets. As shown in Table A2, our Internal Confidence can consistently outperform other baselines on the SimpleQA and MuSiQue tasks. This validates that our training-free method, while not optimal, can obtain very competitive performance without any training samples.
>
> Regarding the optimal centers, we have added Section C.5 to analyze how the center changes across models and datasets.

---

> ### Author Response · Authors · 2025-11-22
> **response (4/5)**
>
> **Q7: Baseline comparisons are limited to adaptations of answer-level methods**
>
> > with no comparison to other query-level uncertainty approaches or simpler heuristics like maximum token probability of the query.
>
> **Answer:**
>
> In the original submission, we did include a couple of heuristics, such as Max(−log p) and Min-K entropy.  For example, Max(−log p) is a baseline that uses the most uncertain token in the query. The results of these baselines are shown in Table 1 and Table A2.
>
> **Q8: Code availability isn't mentioned, and specific prompts and model configurations lack complete documentation, hampering reproducibility and follow-up work.**
>
>
> **Answer:**
>
> We have uploaded our code, which contains the implementation of our Internal Confidence and all the baselines in this manuscript. You can find prompts and model configurations in the `ql_uncertainty` folder.
>
> **Q9: Why is P(YES) specifically the right formulation? Have you tried other self-evaluations (e.g., confidence level on a 1-10 scale)?**
>
> **Answer:**
>
> LLMs can self-evaluate whether they know the answer to a question without reference to any specific proposed answer [1], which indicates that LLMs possess an internal mechanism for assessing the correctness of their outputs.  At the same time, a recent study indicates that answerable and unanswerable questions are also linearly separable in hidden states [2].
>
> Building on this observation, P(Yes) can be regarded as a training-free linear probe to distinguish known and unknown queries.
>
> We agree that there will be other internal signals that can be used for self-evaluation. To the best of our knowledge, we are the first to propose the concept of query-level uncertainty. P(Yes) is not the absolute right formulation, but it is a strong baseline that can be generalized to different LLMs and tasks.
>
> [1] Kadavath et al. Language models (mostly) know what they know
>
> [2] Slobodkin et al. The Curious Case of Hallucinatory (Un)answerability: Finding Truths in the Hidden States of Over-Confident Large Language Models
>
> **Q10: How does the method perform on adversarial inputs where models give confident wrong answers?**
>
> **Answer:**
>
> Our method uses the model’s own internal confidence signal. Therefore, in truly adversarial cases where the model is both wrong and overconfident, the Internal Confidence score will also be high. However, in practice, our Internal Confidence can mitigate this overconfidence.
> In Table 1 and Table A2, we have shown the calibration results (Expected Calibration Error) across different models and tasks. We see that our method consistently has a lower error compared to other baselines.
>
> **Q11: Can you provide correlation analysis between Internal Confidence scores and actual greedy decoding accuracy?**
>
> **Answer:**
>
> Yes, we have demonstrated this correlation analysis in Figure A1. This experiment compares the distributions of Internal Confidence scores for known (green) and unknown (blue) queries across three datasets.  The results reveal that Internal Confidence tends to assign higher values to known queries and lower values to unknown queries, which is suitable for distinguishing the two groups. Specifically, on TriviaQA, the separation is mild with noticeable overlap. On SciQ, the known queries concentrate near 1.0, while unknown queries spread toward lower scores, and on GSM8K, the distinction is the clearest, with known queries clustered in the high-confidence region (0.8–0.9) and unknown queries shifted leftward.
>
> **Q12: How sensitive are results to the specific threshold used for deciding "know/don't know"?**
>
> **Answer:**
>
> In our evaluation, we report an AUC-like metric, which measures the ranking quality of uncertainty scores across all possible thresholds. For example, in Figure 6, we show the performance-cost curves across all thresholds. Therefore, end-users can make trade-offs according to these results.  If high accuracy is required, the threshold can be set to 0.6 for the balance between cost and accuracy. If the inference speed matters, users can choose a threshold that falls in the "trade-off" region according to their specific needs.

---

> ### Author Response · Authors · 2025-11-22
> **response (5/5)**
>
> **Q13: For the decision center analysis (Figure 3b), why does performance peak at h_5^(27) rather than final layer?**
>
> **Answer:**
>
> Based on prior studies, LLMs process and represent information in a hierarchical manner [1,2]. The early layers are primarily responsible for extracting low-level features, while the middle layers begin to integrate this information, forming more complex semantic representations. The late layers are typically dedicated to generating the final output. [3] points out that the factual knowledge is mainly stored in the middle layers. These findings are aligned with our observation: the middle and late layers are more responsible for knowledge-intensive tasks, e.g., knowledge boundary evaluation, rather than the final layer.
>
> [1] Geva et al. Transformer Feed-Forward Layers Build Predictions by Promoting Concepts in the Vocabulary Space.
>
> [2] Wendler et al. Do Llamas Work in English? On the Latent Language of Multilingual Transformers
>
> [3] Identifying Query-Relevant Neurons in Large Language Models for Long-Form Texts
>
> **Q14: Have you tested on models with different architectures (MoE, retrieval-augmented)?**
>
> **Answer:**
>
> Other architectures, such as MoE and multimodal models, are also important. We consider expanding our query-level uncertainty to other architectures in the future.

---

### Official Review · Reviewer_wAg8 · 2025-10-30

**Soundness:** 3
**Presentation:** 3
**Contribution:** 3
**Rating:** 6
**Confidence:** 3

**Summary:**

This paper proposes Query-Level Uncertainty (QLU) for LLMs: a training-free method that estimates whether the model is likely to answer a query correctly before any generation. The core signal Internal Confidence (IC) aggregates self-evaluation logits (YES/NO) across layers $\times$ tokens using a locality-aware attenuation around a fixed decision center (last token, last layer). Experiments on factual QA (TriviaQA, SciQ) and math reasoning (GSM8K) across multiple open models show that IC outperforms post-hoc, answer-dependent baselines in ranking “known vs unknown” queries, while being faster.

**Strengths:**

* Framing “knowledge boundary” as pre-answer uncertainty is useful for agentic pipelines (RAG, slow-thinking, model cascades).
* Training-free and fast. Single forward pass speed largely independent of answer length, high leverage for long-form tasks and tool-heavy agents.
* IC uses only standard hidden states + unembedding, attenuation over layers and tokens captures the information where the “can I answer?” signal concentrates.
* Fixed decision center and a single locality hyper-parameter yield out-of-the-box generalization without per-dataset tuning.
* The paper demonstrates adaptive inference use cases: gating RAG and model cascading to reduce cost at similar accuracy.

**Weaknesses:**

1. Operational definition of “knowledge boundary.” The binary label is tied to greedy decoding success. This could confuse capability with decoding heuristics and underestimate the success of non-greedy decoding. Sensitivity analysis of decoding strategies would help strengthen these claims.
2. IC requires hidden states and full model access. Many production black-box APIs don’t expose them.
3. The method fixes the decision center (last-layer, last-token) and selects a single decay schedule. A more complete ablation including alternative centers, learned decay parameters, or token-type weighting, would reveal the robustness–complexity frontier.
4. Baselines could include some selective prediction and abstention methods (e.g., calibrated confidence, dedicated refuse-to-answer training). This will allow IC to be compared with the strongest alternatives.

**Questions:**

Please refer to the previous weaknesses.

---

> ### Author Response · Authors · 2025-11-22
> **response (1/2)**
>
> Dear Reviewer wAg8,
>
> We thank you for your positive comments and constructive feedback. We appreciate the time you invested in reviewing our paper.
> We are pleased that you find our pre-generation uncertainty useful for agentic pipelines and that our training-free, fast approach is of interest. The manuscript has been revised according to your feedback.
>
> **Q1: Operational definition of “knowledge boundary.**
>
> > Operational definition of “knowledge boundary.” The binary label is tied to greedy decoding success. This could confuse capability with decoding heuristics and underestimate the success of non-greedy decoding. Sensitivity analysis of decoding strategies would help strengthen these claims.
>
> **Answer:**
>
> We have clarified this in the revised manuscript (see the blue texts in Section 3.1).
>
> In the original submission, we emphasized that the setting of greedy decoding serves as a heuristic indicator of internal knowledge, rather than an absolute measure. *"While greedy decoding ensures deterministic measurement, it may not always reflect the optimal
> performance of a model~\citep{song2024good}, as alternative decoding strategies like beam search may elicit a better answer.
> Therefore, this pragmatic framework serves as a heuristic indicator of internal knowledge, rather than an absolute measure.
> We use this standard to evaluate the estimated query-level uncertainty, i.e., a lower uncertainty indicates a model is more likely to output the correct answer. "*
>
> To clarify this, we added further description in this section (blue texts) to explain why we stick with greedy decoding:
> * Efficiency. Our method treats a successful greedy decode as a signal that the model knows how to answer, which is a fast proxy. In contrast, non-greedy decoding requires the configuration of beam sizes, probability thresholds, and sampling numbers, which complicates both the definition of the knowledge boundary and the assessment cost. How to define a correct answer in a non-greedy decoding setting is tricky.
> * Reproducibility. Greedy decoding outputs a single deterministic output for a given input, which offers a stable and reproducible baseline and benchmark.
>
> At the same time, we have added experiments to study the query-level uncertainty under a non-greedy decoding setting.
> Specifically, we enable sampling, set the beam size to 5, and use a temperature of 1.0. We assess different query-level uncertainty methods on Llama-8B using the TriviaQA dataset.
> For each query, we prompt the model to generate 10 answers. If the model produces the correct answer in more than 5 out of the 10 attempts, we consider the query to fall within the model’s knowledge boundary.
> The results are shown in the Table below. We can see that our Internal Confidence can identify known and unknown queries better than other competitors, and the scores (also the ranking of each method) are highly similar to the results under greedy decoding. These consistent findings confirm that our greedy decoding setting is a simple yet effective and robust way to measure the knowledge boundary.
>
>
> | Method                 | AUC ↑ | PRR ↑ | ECE ↓ |
> |------------------------|-------|-------|-------|
> | Max(– log p)           | 54.4  | 11.6  | ---   |
> | Predictive Entropy     | 57.6  | 10.8  | ---   |
> | Min-K Entropy          | 59.1  | 19.1  | ---   |
> | Attentional Entropy    | 61.2  | 23.5  | ---   |
> | Perplexity             | 61.0  | 24.8  | ---   |
> | P(YES) (top right)     | 57.1  | 14.0  | 26.0  |
> | P(YES) (naive avg)     | 66.3  | 30.0  | **18.1** |
> | Internal Confidence    | **71.7** | **45.7** | 19.9 |
>
> **Q2: IC requires hidden states and full model access.**
>
> > Many production black-box APIs don’t expose them.
>
> **Answer:**
>
> Completely black-box APIs generally do not provide internal hidden states, so our method is not applicable in this case.
> Commercial LLM hosts can apply our method for adaptive inference before model outputs are exposed to end users.
> On the other hand, our method remains applicable to any API that provides token-level probabilities. For instance, the table below shows that our Internal Confidence offers a working solution when the API exposes the last-layer token probabilities.
>
> | Method                            | TriviaQA | SciQ | GSM8K |
> |-----------------------------------|----------|------|--------|
> | Internal Confidence (all layers) | 68.7     | 58.1 | 65.7   |
> | Internal Confidence (last layer) | 67.9     | 56.3 | 65.3   |

---

> ### Author Response · Authors · 2025-11-22
> **response (2/2)**
>
> **Q3: The method fixes the decision center (last-layer, last-token) and selects a single decay schedule. .**
>
> > A more complete ablation including alternative centers, learned decay parameters, or token-type weighting, would reveal the robustness–complexity frontier.
>
>
> **Answer:**
>
> We have added experiments to study the learned decision center across different models and tasks. Figure A4-6 in the appendix show the locations of decision centers. We can observe that the center tends to appear at the top right place for TriviaQA and SciQ, while the math reasoning task of GSM8K has a distinct behavior. The center is located in the lower layers. Although the current default center (top right) is sub-optimal, it offers a training-free and strong baseline, which can be generalized across different applications.
>
> We have studied the robustness of our method about the decay parameters and token weights (locality). The results are shown in Figure A3 in the appendix. These weights have a non-trivial effect on the performance, and their optimal value varies slightly by model and dataset. Phi-3.8B and Qwen-14B benefit more clearly from tuning locality, while Llama-8B appears more robust to changes. Overall, high locality values often yield competitive or optimal performance.
>
> **Q4: Baselines**
>
> > Baselines could include some selective prediction and abstention methods (e.g., calibrated confidence, dedicated refuse-to-answer training). This will allow IC to be compared with the strongest alternatives.
>
> **Answer:**
>
> We have included more answer-level uncertainty approaches, as shown in the table below. Our internal confidence is able to achieve competitive performance, especially for long-form tasks such as GSM8K, while being extremely fast.
>
> Additionally, we test query-level uncertainty on three new and more complex datasets (see Table A2 in the appendix), which confirms the effectiveness of our method.
>
> | Method               | TriviaQA AUC | TriviaQA Time | TriviaQA Speedup | SciQ AUC | SciQ Time | SciQ Speedup | GSM8K AUC | GSM8K Time | GSM8K Speedup |
> |----------------------|--------------|----------------|-------------------|----------|-----------|---------------|-----------|------------|----------------|
> | CCP                  | 73.3         | 37.6           | 188X              | 63.1     | 48.9      | 245X          | 64.2      | 61.7       | 206X           |
> | MSP                  | 76.9         | 2.49           | 13X               | **70.3** | 3.85      | 19X           | **68.5**  | 25.1       | 84X            |
> | EigValLaplacian      | **78.1**     | 12.4           | 62X               | 65.7     | 23.6      | 118X          | 56.7      | 153.8      | 512X           |
> | Lexical Similarity   | 77.2         | 140.3          | 702X              | 68.7     | 165.7     | 829X          | 62.4      | 22.3       | 74X            |
> | DegMat               | 75.0         | 12.2           | 61X               | 63.1     | 25.1      | 125X          | 62.1      | 153.9      | 513X           |
> | TokenSAR             | 78.0         | 17.4           | 87X               | 50.1     | 27.1      | 135X          | 62.2      | 189.5      | 632X           |
> | Internal Confidence  | 71.9         | **0.2**        | —                 | 62.6     | **0.2**   | —             | 66.8      | **0.3**    | —              |

---

### Official Review · Reviewer_ZpJR · 2025-10-31

**Soundness:** 2
**Presentation:** 3
**Contribution:** 3
**Rating:** 4
**Confidence:** 5

**Summary:**

The paper proposes a training-free, pre-generation measure of query-level uncertainty called Internal Confidence (IC). Instead of generating an answer and then judging it, IC probes the model’s hidden states at many token positions nnn and layers lll using the model’s unembedding rows for {YES, NO}. These probe scores are then aggregated with decay weights around a “decision center” (empirically near the last token & last layer) to produce a single confidence score. The paper motivates the center by showing that discrimination between known/unknown queries often peaks at an interior token×layer location, then ensembles nearby positions via an “attenuated encoding”. Experiments on factual QA and math reasoning claim better AUC/PRR and calibration (ECE) than query-level baselines, and show major speedups versus answer-level methods (e.g., SAR), enabling applications like cost-aware RAG routing and model cascading.

**Strengths:**

Pre-generation, single-pass signal. IC avoids generating answers and extra prompts; it’s computed from internal states with one forward pass, which is appealing for latency/cost.

Center-weighted ensembling across tokens×layers. The “decision center” idea is well-motivated by heatmaps showing the best separator isn’t always exactly the last position; the attenuated average reduces variance while keeping locality.

Practical routing use-cases. Clear demonstrations for RAG triggering and small→large model cascading with thresholding on IC (trade-off/optimal regions).

**Weaknesses:**

1) Baselines. It is puzzling that the baselines differ between Tables 1 and 2; they should be consistent to allow a fair comparison of self-knowledge and efficiency across models and tasks. Moreover, the state-of-the-art claim cannot be substantiated without including a broader set of strong baselines. Following the recent TACL benchmark on uncertainty quantification [1] , at minimum the evaluation should cover half of these baselines (that outperform SAR as a top baseline from the paper): CCP, Maximum Sequence Probability (MSP), EigValLaplacian, Lexical Similarity (ROUGE-L), DegMat, TokenSAR, representative verbalized-uncertainty methods, and the recently proposed RAUQ. For references and straightforward replication, please see LM-Polygraph.


2) Datasets. The current dataset selection provides limited coverage of multi-hop and genuinely challenging questions. Are the proposed methods generalizable to such settings? TriviaQA and SciQ contain many single-fact lookups and shallow cues, which can inflate confidence estimates. Please consider adding SimpleQA [2]  (challenging for LLMs and less likely to be contaminated by pretraining), TruthfulQA[3]  (tests suppression of “popular but false” answers), and multi-hop benchmarks such as MuSiQue[4] and 2WikiMultihopQA [5].

        For mathematics, GSM8K largely features short, templated chains, where uncertainty may     correlate with sequence length or final-step carry errors rather than true multi-step epistemic gaps. We recommend evaluating on GSM8K-Hard[6], which require longer chains and introduce more realistic failure modes.

3) Adaptive RAG. What you refer to as “efficient RAG” is a fast-moving area with a substantial body of adaptive RAG/adaptive retrieval baselines. To substantiate the contribution, it’s important to compare against representative prior work—for example DRAGIN [7], FLARE [8], SEAKR[9],  AdaptiveRAG[10], and recent approaches that leverage uncertainty [11]  and LLM-independent features [12]. Including these would better situate your method within the existing literature.

4) Access assumptions. IC needs logits / unembedding and hidden states. That’s fine for open-weights models, but not available for black-box APIs, where answer-level methods (e.g., P(True)) might be more deployable. Can the method work if black-box models provide access to the last layer, some per-output-token log-probabilities?


5) Definition of “knowledge boundary.” The authors evaluate answerability under greedy decoding as a proxy; this can misclassify queries where non-greedy decoding would succeed, so “known vs unknown” labels are not absolute.


Typos:: Notation mismatch: “w = 1.0” vs Eq. (4). In §4.5 the text says “vary the w in Equation 4 … default w = 1.0 (Locality ≈ 0.7).” But Eq. (4) defines the attenuation with parameter α controlling locality. This reads like a symbol mix-up; likely they meant α = 1.0. Please unify notation.


[1] Vashurin, R., Fadeeva, E., Vazhentsev, A., Rvanova, L., Vasilev, D., Tsvigun, A., ... & Shelmanov, A. (2025). Benchmarking uncertainty quantification methods for large language models with lm-polygraph. Transactions of the Association for Computational Linguistics, 13, 220-248.

[2] Wei, J., Karina, N., Chung, H. W., Jiao, Y. J., Papay, S., Glaese, A., ... & Fedus, W. (2024). Measuring short-form factuality in large language models. arXiv preprint arXiv:2411.04368.

[3] Lin, S., Hilton, J., & Evans, O. (2021). Truthfulqa: Measuring how models mimic human falsehoods. arXiv preprint arXiv:2109.07958.

[4] Trivedi, H., Balasubramanian, N., Khot, T., & Sabharwal, A. (2022). ♫ MuSiQue: Multihop Questions via Single-hop Question Composition. Transactions of the Association for Computational Linguistics, 10, 539-554.

[5] Ho, X., Nguyen, A. K. D., Sugawara, S., & Aizawa, A. (2020). Constructing a multi-hop qa dataset for comprehensive evaluation of reasoning steps. arXiv preprint arXiv:2011.01060.

[6] Gao, L., Madaan, A., Zhou, S., Alon, U., Liu, P., Yang, Y., ... & Neubig, G. (2023, July). Pal: Program-aided language models. In International Conference on Machine Learning (pp. 10764-10799). PMLR.

[7] Su, W., Tang, Y., Ai, Q., Wu, Z., & Liu, Y. (2024). DRAGIN: dynamic retrieval augmented generation based on the information needs of large language models. arXiv preprint arXiv:2403.10081.

[8] Jiang, Z., Xu, F. F., Gao, L., Sun, Z., Liu, Q., Dwivedi-Yu, J., ... & Neubig, G. (2023, December). Active retrieval augmented generation. In Proceedings of the 2023 Conference on Empirical Methods in Natural Language Processing (pp. 7969-7992).

[9]  Yao, Z., Qi, W., Pan, L., Cao, S., Hu, L., Liu, W., ... & Li, J. (2024). Seakr: Self-aware knowledge retrieval for adaptive retrieval augmented generation. arXiv preprint arXiv:2406.19215.

[10] Jeong, S., Baek, J., Cho, S., Hwang, S. J., & Park, J. C. (2024). Adaptive-rag: Learning to adapt retrieval-augmented large language models through question complexity. arXiv preprint arXiv:2403.14403.

[11] Moskvoretskii, V., Lysyuk, M., Salnikov, M., Ivanov, N., Pletenev, S., Galimzianova, D., ... & Panchenko, A. (2025). Adaptive retrieval without self-knowledge? bringing uncertainty back home. arXiv preprint arXiv:2501.12835.

[12] Marina, M., Ivanov, N., Pletenev, S., Salnikov, M., Galimzianova, D., Krayko, N., ... & Moskvoretskii, V. (2025). LLM-Independent Adaptive RAG: Let the Question Speak for Itself. arXiv preprint arXiv:2505.04253.

**Questions:**

I appreciate the training-free setup and the promise of OOD-robust, pre-generation uncertainty. However, the evidence feels incomplete primarily because key, state-of-the-art baselines are missing. If the authors add stronger baselines (e.g., CCP, MSP, TokenSAR, DegMat/EigVal, RAUQ, robust verbalized-uncertainty) and evaluate on harder datasets, I could  revise my view. I also recommend reframing the main claim around efficiency (with explicit token/latency budgets) and presenting accuracy as on-par with SOTA where that is borne out.

---

> ### Author Response · Authors · 2025-11-22
> **response (1/3)**
>
> Dear Reviewer ZpJR,
>
> Thank you for your insightful and actionable feedback. We appreciate the time you invested in reviewing our manuscript. We are also encouraged that you find our proposed method well-motivated, appealing, and practical. The required baselines and datasets have been added, and we rephrased the corresponding claims. We hope we have addressed your concerns.
>
> **Q1: Baselines**
>
> > Baselines. It is puzzling that the baselines differ between Tables 1 and 2; they should be consistent to allow a fair comparison of self-knowledge and efficiency across models and tasks. Moreover, the state-of-the-art claim cannot be substantiated without including a broader set of strong baselines. Following the recent TACL benchmark on uncertainty quantification [1] , at minimum the evaluation should cover half of these baselines (that outperform SAR as a top baseline from the paper): CCP, Maximum Sequence Probability (MSP), EigValLaplacian, Lexical Similarity (ROUGE-L), DegMat, TokenSAR, representative verbalized-uncertainty methods, and the recently proposed RAUQ. For references and straightforward replication, please see LM-Polygraph.
>
> **Answer:**
>
> You are right. We are not aiming to propose a state-of-the-art uncertainty approach. The core motivation is to develop a fast and cheap method that can detect the knowledge boundaries of a model, which allows us to perform adaptive inference to reduce the computational cost. We have compared our Internal Confidence to some answer-level baselines, such as SAR and Lexical Similarity, as shown in Table 2 and Table A1.
> Following your suggestions, we conducted experiments to include stronger baselines.  The results are shown in the table below. We observe that our Internal Confidence is able to achieve competitive performance while being much faster, especially for mathematical reasoning. According to this, we rephrased our main claim in Line 425 (blue texts): *"These results demonstrate that Internal Confidence achieves competitive performance compared to answer-level uncertainty approaches while being orders of magnitude faster"*.
>
>
>
> | Method               | TriviaQA AUC | TriviaQA Time | TriviaQA Speedup | SciQ AUC | SciQ Time | SciQ Speedup | GSM8K AUC | GSM8K Time | GSM8K Speedup |
> |----------------------|--------------|----------------|-------------------|----------|-----------|---------------|-----------|------------|----------------|
> | CCP                  | 73.3         | 37.6           | 188X              | 63.1     | 48.9      | 245X          | 64.2      | 61.7       | 206X           |
> | MSP                  | 76.9         | 2.49           | 13X               | **70.3** | 3.85      | 19X           | **68.5**  | 25.1       | 84X            |
> | EigValLaplacian      | **78.1**     | 12.4           | 62X               | 65.7     | 23.6      | 118X          | 56.7      | 153.8      | 512X           |
> | Lexical Similarity   | 77.2         | 140.3          | 702X              | 68.7     | 165.7     | 829X          | 62.4      | 22.3       | 74X            |
> | DegMat               | 75.0         | 12.2           | 61X               | 63.1     | 25.1      | 125X          | 62.1      | 153.9      | 513X           |
> | TokenSAR             | 78.0         | 17.4           | 87X               | 50.1     | 27.1      | 135X          | 62.2      | 189.5      | 632X           |
> | Internal Confidence  | 71.9         | **0.2**        | —                 | 62.6     | **0.2**   | —             | 66.8      | **0.3**    | —              |

---

> ### Author Response · Authors · 2025-11-22
> **response (2/3)**
>
> **Q2: Datasets**
>
> > The current dataset selection provides limited coverage of multi-hop and genuinely challenging questions. Are the proposed methods generalizable to such settings? TriviaQA and SciQ contain many single-fact lookups and shallow cues, which can inflate confidence estimates. Please consider adding SimpleQA [2] (challenging for LLMs and less likely to be contaminated by pretraining), TruthfulQA[3] (tests suppression of “popular but false” answers), and multi-hop benchmarks such as MuSiQue[4] and 2WikiMultihopQA [5].
>
> **Answer:**
>
> Following your suggestion, we conducted additional experiments to include more challenging datasets: SimpleQA, TruthfulQA, and MuSiQue, which aim to offer less contaminated and multi-hop benchmarks. The details and results are shown in Section C.4 in the manuscript. We also report the results of phi below due to the space limit. We can observe that our Internal Confidence still outperforms other query-level baselines on more challenging datasets.
>
> | Model         | Method                       | SimpleQA AUC | PRR | ECE | MuSiQue AUC | PRR | ECE | TruthfulQA AUC | TruthfulQA PRR | TruthfulQA ECE | Avg AUC | Avg PRR | Avg ECE |
> |---------------|------------------------------|-------------:|-------------:|-------------:|-------------:|-------------:|-------------:|---------------:|----------------|---------------:|--------:|--------:|--------:|
> | **Phi-3.8B**  | Max (-log p)                 | 50.5        | 5.5          | –            | 53.2         | 4.7          | –            | 50.3           | –1.3            | –             | 51.3    | 3.0     | –      |
> |               | Predictive Entropy           | 54.0        | 5.9          | –            | **65.6**      | 29.6         | –            | 55.7           | **15.6**         | –             | 58.4    | 17.0    | –      |
> |               | Min-K Entropy                | 53.8        | 13.2         | –            | 59.9         | 21.9         | –            | 55.1           | 13.9            | –             | 56.3    | 16.3    | –      |
> |               | Attentional Entropy          | 50.1        | 5.1          | –            | 54.7         | 11.6         | –            | 52.4           | 8.3             | –             | 52.4    | 8.3     | –      |
> |               | Perplexity                   | 52.4        | 5.8          | –            | 55.2         | 7.1          | –            | 54.1           | 3.8             | –             | 53.9    | 5.6     | –      |
> |               | P(Yes)  (top right)         | **61.3**     | 17.8         | 18.8         | 65.4         | 29.8         | 9.5          | 39.6           | –16.2           | 27.1          | 55.4    | 10.5    | **18.5**|
> |               | P(Yes)  (naive avg)         | 59.8        | 17.8         | 69.9         | 65.5         | 29.5         | 63.2         | 49.3           | –2.0            | **25.5**       | 58.2    | 15.1    | 52.9   |
> |               | Internal Confidence          | 61.2        | **26.1**     | **18.2**     | 65.5         | **30.2**     | **9.3**       | **56.4**        | 13.2            | 40.7          | **61.0** | **23.2**| 22.7   |
>
> **Q3: Adaptive RAG.**
>
> > What you refer to as “efficient RAG” is a fast-moving area with a substantial body of adaptive RAG/adaptive retrieval baselines. To substantiate the contribution, it’s important to compare against representative prior work—for example DRAGIN [7], FLARE [8], SEAKR[9], AdaptiveRAG[10], and recent approaches that leverage uncertainty [11] and LLM-independent features [12]. Including these would better situate your method within the existing literature.
>
> **Answer:**
>
> These adaptive RAG methods are correlated with our efficient RAG. We have added a discussion in Section 4.4.
> *"Existing studies have explored adaptive RAG through learned classifiers (Jeong et al. 2024; Marina et al. 2025) and answer-level uncertainty approaches (Jiang et al. 2023); Su et al. 2024; Yao et al. 2025; Moskvoretskii et al. 2025), which actively decide whether and when to retrieve documents. However, these approaches require training samples or generating answers to measure the uncertainty. In contrast, our Internal Confidence method is training-free and significantly faster than answer-level approaches, as shown in Table 2., which can serve as a potentially efficient way to guide adaptive RAG."*

---

> ### Author Response · Authors · 2025-11-22
> **response (3/3)**
>
> **Q4: Access assumptions.**
>
> > IC needs logits / unembedding and hidden states. That’s fine for open-weights models, but not available for black-box APIs, where answer-level methods (e.g., P(True)) might be more deployable. Can the method work if black-box models provide access to the last layer, some per-output-token log-probabilities?
>
> **Answer:**
> Our method can be applied to black-box models if the last-layer token probability is accessible. We have tested the performance if only the last layer is visible, and the AUC results of Llama-8B are shown in the table below. The findings reveal that all-layer Internal Confidence provides better scores due to the rich information across layers and tokens, but last-layer internal confidence does not degrade much, and it offers a feasible approach when the model is not fully white-box.
>
> | Method                            | TriviaQA | SciQ | GSM8K |
> |-----------------------------------|----------|------|--------|
> | Internal Confidence (all layers) | 68.7     | 58.1 | 65.7   |
> | Internal Confidence (last layer) | 67.9     | 56.3 | 65.3   |
>
>
> **Q5: Definition of “knowledge boundary."**
>
> >  The authors evaluate answerability under greedy decoding as a proxy; this can misclassify queries where non-greedy decoding would succeed, so “known vs unknown” labels are not absolute.
>
> **Answer:**
>
> We have clarified this in the revised manuscript (see the blue texts in Section 3.1).
>
> In the original submission, we emphasized that the setting of greedy decoding serves as a heuristic indicator of internal knowledge, rather than an absolute measure. *"While greedy decoding ensures deterministic measurement, it may not always reflect the optimal performance of a model~\citep{song2024good}, as alternative decoding strategies like beam search may elicit a better answer.
> Therefore, this pragmatic framework serves as a heuristic indicator of internal knowledge, rather than an absolute measure.
> We use this standard to evaluate the estimated query-level uncertainty, i.e., a lower uncertainty indicates a model is more likely to output the correct answer. "*
>
> To clarify this, we added further description in this section (blue texts) to explain why we stick with greedy decoding:
> * Efficiency. Our method treats a successful greedy decode as a signal that the model knows how to answer, which is a fast proxy. In contrast, non-greedy decoding requires the configuration of beam sizes, probability thresholds, and sampling numbers, which complicates both the definition of the knowledge boundary and the assessment cost. How to define a correct answer in a non-greedy decoding setting is tricky.
> * Reproducibility. Greedy decoding outputs a single deterministic output for a given input, which offers a stable and reproducible baseline and benchmark.
>
> At the same time, we have added experiments to study the query-level uncertainty under a non-greedy decoding setting.
> Specifically, we enable sampling, set the beam size to 5, and use a temperature of 1.0. We assess different query-level uncertainty methods on Llama-8B using the TriviaQA dataset.
> For each query, we prompt the model to generate 10 answers. If the model produces the correct answer in more than 5 out of the 10 attempts, we consider the query to fall within the model’s knowledge boundary.
> The results are shown in the Table below. We can see that our Internal Confidence can identify known and unknown queries better than other competitors, and the scores (also the ranking of each method) are highly similar to the results under greedy decoding. These consistent findings confirm that our greedy decoding setting is a simple yet effective and robust way to measure the knowledge boundary.
>
> | Method                 | AUC ↑ | PRR ↑ | ECE ↓ |
> |------------------------|-------|-------|-------|
> | Max(– log p)           | 54.4  | 11.6  | ---   |
> | Predictive Entropy     | 57.6  | 10.8  | ---   |
> | Min-K Entropy          | 59.1  | 19.1  | ---   |
> | Attentional Entropy    | 61.2  | 23.5  | ---   |
> | Perplexity             | 61.0  | 24.8  | ---   |
> | P(YES) (top right)     | 57.1  | 14.0  | 26.0  |
> | P(YES) (naive avg)     | 66.3  | 30.0  | **18.1** |
> | Internal Confidence    | **71.7** | **45.7** | 19.9 |
>
> **Q6: Typos**
>
> > Notation mismatch: “w = 1.0” vs Eq. (4). In §4.5 the text says “vary the w in Equation 4 … default w = 1.0 (Locality ≈ 0.7).” But Eq. (4) defines the attenuation with parameter α controlling locality. This reads like a symbol mix-up; likely they meant α = 1.0. Please unify notation**
>
> **Answer:**
>
> Thanks for spotting this typo. We have corrected it in the revised manuscript.

---

> > ### Comment · Reviewer_ZpJR · 2025-11-26
> > **Reviewer Answer**
> >
> > Thank you for your responses; most of my concerns have been addressed, and I have revised my score accordingly.
> >
> > However, it appears that some simple baselines (such as MSP, which is far less computationally expensive) outperform your method substantially (by about 8 points) on SciQ, so it can hardly be described as “competitive with SOTA.” Moreover, these new baselines are missing from the additional comparisons you provide (your responses 2/3 and 3/3), which still makes a fair evaluation between your method and other methods challenging.

---

> > > ### Author Response · Authors · 2025-12-01
> > >
> > > Dear Reviewer ZpJR,
> > >
> > > We would like to thank you for raising the score from 4->6. We appreciate your valuable and actionable suggestions, which have helped us improve our manuscript.
> > >
> > > **Q: However, it appears that some simple baselines (such as MSP, which is far less computationally expensive) outperform your method substantially (by about 8 points) on SciQ, so it can hardly be described as “competitive with SOTA.” Moreover, these new baselines are missing from the additional comparisons you provide (your responses 2/3 and 3/3), which still makes a fair evaluation between your method and other methods challenging.**
> > >
> > > **Answer:**
> > >
> > > Thanks for pointing this out.
> > >
> > > We agree that some answer-level baselines perform better on SciQ. However, these methods require generating full answers, which makes them significantly slower than our proposed approach. In contrast, Internal Confidence operates at the query level and achieves up to a 25× speedup with only a modest 1.7-point reduction in AUC, as shown in Table 2. From this perspective, we believe it is fair to state that *"These results demonstrate that Internal Confidence achieves competitive performances compared to answer-level uncertainty approaches while being extremely faster, which
> > > can be a practical choice for tasks requiring longer and more complex answers."*
> > >
> > > We emphasize the low computational cost and reasonable overall performance of our method, which remains competitive, particularly for tasks requiring longer and more complex answers. The corresponding claim has been slightly revised to reflect this (line 427).

---

### Author Response · Authors · 2025-12-02
**Global Response (1/2)**

Dear Reviewers and AC,

We would like to thank you for your insightful feedback and constructive suggestions, which have greatly improved the clarity and soundness of our manuscript. We appreciate the efforts you spent reviewing our work.  Below, we summarize the main concerns and our responses.

**Q1: Datasets. The current dataset selection provides limited coverage of multi-hop and genuinely challenging questions (Reviewer ZpJR). The evaluation scope is narrow and raises generalization concerns (Reviewer hjve);**

**Answer:**

Following the suggestion of Reviewer ZpJR, we conduct experiments on three additional datasets that are more difficult:
* SimpleQA. This is a benchmark that evaluates the ability of language models to answer short, fact-relevant questions, which is less likely to be contaminated by the pre-training stage.
* MuSiQue. This is a dataset that
requires proper multihop reasoning, which is more difficult and  harder to cheat via disconnected reasoning.
* TruthfulQA. This is a benchmark to measure whether a language model is truthful in generating answers to questions.

The details and results are shown in Section C.4 in the manuscript. We also report the results of Phi-3.8B below due to the space limit. We can observe that our Internal Confidence still outperforms other query-level baselines on more challenging datasets.

| Model         | Method                       | SimpleQA AUC | PRR | ECE | MuSiQue AUC | PRR | ECE | TruthfulQA AUC | TruthfulQA PRR | TruthfulQA ECE | Avg AUC | Avg PRR | Avg ECE |
|---------------|------------------------------|-------------:|-------------:|-------------:|-------------:|-------------:|-------------:|---------------:|----------------|---------------:|--------:|--------:|--------:|
| **Phi-3.8B**  | Max (-log p)                 | 50.5        | 5.5          | –            | 53.2         | 4.7          | –            | 50.3           | –1.3            | –             | 51.3    | 3.0     | –      |
|               | Predictive Entropy           | 54.0        | 5.9          | –            | **65.6**      | 29.6         | –            | 55.7           | **15.6**         | –             | 58.4    | 17.0    | –      |
|               | Min-K Entropy                | 53.8        | 13.2         | –            | 59.9         | 21.9         | –            | 55.1           | 13.9            | –             | 56.3    | 16.3    | –      |
|               | Attentional Entropy          | 50.1        | 5.1          | –            | 54.7         | 11.6         | –            | 52.4           | 8.3             | –             | 52.4    | 8.3     | –      |
|               | Perplexity                   | 52.4        | 5.8          | –            | 55.2         | 7.1          | –            | 54.1           | 3.8             | –             | 53.9    | 5.6     | –      |
|               | P(Yes)  (top right)         | **61.3**     | 17.8         | 18.8         | 65.4         | 29.8         | 9.5          | 39.6           | –16.2           | 27.1          | 55.4    | 10.5    | **18.5**|
|               | P(Yes)  (naive avg)         | 59.8        | 17.8         | 69.9         | 65.5         | 29.5         | 63.2         | 49.3           | –2.0            | **25.5**       | 58.2    | 15.1    | 52.9   |
|               | Internal Confidence          | 61.2        | **26.1**     | **18.2**     | 65.5         | **30.2**     | **9.3**       | **56.4**        | 13.2            | 40.7          | **61.0** | **23.2**| 22.7   |

---

### Author Response · Authors · 2025-12-02
**Global Response (2/2)**

**Q2: definition of knowledge boundary.  The authors evaluate answerability under greedy decoding as a proxy; this can misclassify queries where non-greedy decoding would succeed (Reviewer ZpJR, wAg8, hjve, and pkEN)**

**Answer:**

In the original submission, we emphasized that the setting of greedy decoding serves as a practical heuristic indicator of internal knowledge, rather than an absolute measure (see Section 3.1).
The goal is to approximate whether the model possesses the necessary knowledge to answer the query, not to exhaust all possible decoding strategies. This measurement is a fast and reproducible proxy that yields results consistent with those obtained using non-greedy decoding. More importantly, greedy decoding helps adaptive inference to reduce inference cost that sampling-based methods cannot offer.

We have added experiments to study the query-level uncertainty under a non-greedy decoding setting.
The results are shown in the Table below. We can see that the scores (also the ranking of each method) are highly similar to the results under greedy decoding. These consistent findings confirm that our greedy decoding setting is a simple yet effective and robust way to measure the knowledge boundary.

| Method                 | AUC ↑ | PRR ↑ | ECE ↓ |
|------------------------|-------|-------|-------|
| Max(– log p)           | 54.4  | 11.6  | ---   |
| Predictive Entropy     | 57.6  | 10.8  | ---   |
| Min-K Entropy          | 59.1  | 19.1  | ---   |
| Attentional Entropy    | 61.2  | 23.5  | ---   |
| Perplexity             | 61.0  | 24.8  | ---   |
| P(YES) (top right)     | 57.1  | 14.0  | 26.0  |
| P(YES) (naive avg)     | 66.3  | 30.0  | **18.1** |
| Internal Confidence    | **71.7** | **45.7** | 19.9 |

Specifically, in this experiment, we enable sampling, set the beam size to 5, and use a temperature of 1.0. We assess different query-level uncertainty methods on Llama-8B using the TriviaQA dataset.
For each query, we prompt the model to generate 10 answers. If the model produces the correct answer in more than 5 out of the 10 attempts, we consider the query to fall within the model’s knowledge boundary.

**Q3: Access assumptions. IC needs logits / unembedding and hidden states (Reviewer ZpJR); IC requires hidden states and full model access. Many production black-box APIs don’t expose them.(Reviewer wAg8)**

**Answer:**

We agree that our method cannot be applied to fully black-box APIs.

However, our method is feasible for black-box models if the last-layer token probability is accessible. We have tested the performance if only the last layer is visible, and the AUC results of Llama-8B are shown in the table below. The findings reveal that all-layer Internal Confidence provides better scores due to the rich information across layers and tokens, but last-layer internal confidence does not degrade much, and it offers a feasible approach when the model is not fully white-box.

| Method                            | TriviaQA | SciQ | GSM8K |
|-----------------------------------|----------|------|--------|
| Internal Confidence (all layers) | 68.7     | 58.1 | 65.7   |
| Internal Confidence (last layer) | 67.9     | 56.3 | 65.3   |

---

### Meta-Review · Area_Chair_doJQ · 2026-01-06

**Summary:**

The paper proposes a training-free generation-free method for estimating uncertainty / likelihood of failure on a certain query before generating the answer. The reviewers recognize its efficiency, simplicity, robustness and competitive performance. The reviewers also raised concerns about sufficient baseline comparisons and some design choices (such as greedy decoding).

**Reviewer Concerns:**

See below.

**Reviewer Scores:**

Reviewer ZpJR appreciated the method’s efficiency and motivation, but raised several concerns 1) baseline consistency and missing stronger baselines in uncertainty estimation and RAG, 2) non-applicability to closed/API models. I believe concern 2) can’t be a basis for rejection. The reviewers addressed most of these concerns (except for RAG) in their response and the reviewer was going to raise the score 4->6.

Reviewer wAg8 raises the following concerns: 1) dependence on greedy decoding, 2) non-applicability to closed/API models, 3) lack of ablation, 4) lack of baselines. I believe the author rebuttal sufficiently addressed these questions and the reviewer would retain the score 6.

Reviewer hjve raised the following concerns: 1) lack of theoretical grounding, 2) narrow evaluation scope, 3) greedy decoding, 4) limited baselines. I believe that concerns 1) and 2) can be discarded based on the scope of the paper. The authors provided additional datasets and baselines in the rebuttal. I believe that most concerns were addressed by rebuttal and the score would be raised 4->6.

Reviewer pkEN raised concerns about clarity and experimental results. The reviewer indicated they were unlikely not going to increase the score and retained the score 4.

Overall, although the paper is borderline, I believe its strengths outweigh its weaknesses and I recommend an accept.

---

### Decision · Program_Chairs · 2026-01-26

Accept (Poster)